# Long-term measurements of ice nucleating particles at Atmospheric Radiation Measurement (ARM) sites worldwide

Jessie M. Creamean[1], Carson Hume[1], Maria Vazquez[1], Adam Theisen[2]

[1]Department of Atmospheric Science, Colorado State University, Fort Collins, Colorado, 80523, USA

[2]Argonne National Laboratory, Lemont, Illinois, 60439, USA

*Correspondence to*: Jessie M. Creamean (jessie.creamean@colostate.edu)

**Abstract**

Ice nucleating particles (INPs) play a critical role in cloud microphysics and precipitation formation, yet long-term, spatially extensive observational datasets remain limited. Here, we present one of the most comprehensive publicly available datasets of immersion-mode INP concentrations using a single analytical method, generated through the U.S. Department of Energy's (DOE) Atmospheric Radiation Measurement (ARM) user facility. INP filter samples have been collected across a broad range of environments—including agricultural plains, Arctic coastlines, high-elevation mountain sites, marine regions, and urban areas—via fixed observatories, mobile facility deployments, and vertically-resolved tethered balloon system operations. We describe the standardized processing and quality assurance pipeline, from filter collection and processing using the Ice Nucleation Spectrometer to final data products archived on the ARM Data Discovery portal. The dataset includes both total INP concentrations and selectively treated samples, allowing for classification of biological, organic, and inorganic INP types. It features a continuous 5-year record of INP measurements from a central U.S. site, with data collection still ongoing. Seasonal and site-specific differences in INP concentrations are illustrated through intercomparisons at $-10\,°C$ and $-20\,°C$, revealing distinct regional sources and atmospheric drivers. We also outline mechanisms for researchers to access existing data, request additional sample analyses, and propose future field campaigns involving ARM INP measurements. This dataset supports a wide range of scientific applications, from observational and mechanistic studies to model development, and provides critical constraints on aerosol-cloud interactions across diverse atmospheric regimes.

**Short summary**

This study presents a comprehensive, publicly available ice nucleating particles (INP) dataset from the U.S. Department of Energy Atmospheric Radiation Measurement (ARM) user facility across diverse environments, including Arctic, agricultural, urban, marine, and mountainous sites. Samples are collected via fixed and mobile platforms and processed using a standardized pipeline. The dataset supports observational and modelling analyses of seasonal, spatial, and compositional variability in INPs.

## 1 Introduction

The formation and microphysical evolution of cloud droplets and ice crystals are strongly influenced by aerosols acting as cloud condensation nuclei (CCN) and ice nucleating particles (INPs). While INP observations remain sparse compared to other aerosol properties, they are essential for understanding aerosol-cloud interactions and their impacts on cloud microphysics and radiative properties. Immersion freezing—where an INP first acts as a CCN before freezing at temperatures above homogeneous freezing ($-38\,°C$)—is particularly important for mixed-phase cloud formation (Kanji et al., 2017; Knopf and Alpert, 2023).

An aerosol's ability to serve as an INP depends on temperature, vapor saturation with respect to water and ice, and particle properties such as composition (chemical, mineral, or biological), morphology, and size, all of which are linked to its source (Hoose and Möhler, 2012). Known atmospheric INPs include mineral dust, soil dust, sea spray, volcanic ash, black carbon, and a range of biological particles (e.g., bacteria, fungal spores, pollen, algae, lichens, macromolecules) (e.g., Conen et al., 2011; Creamean et al., 2013, 2019; Cziczo et al., 2017; DeMott, 1990; DeMott et al., 2016, 2018c; Hill et al., 2016; Huang et al., 2021; Kaufmann et al., 2016; Levin et al., 2010; McCluskey et al., 2017; O'Sullivan et al., 2014, 2016). Among natural INPs, mineral dust and biological particles are especially important. Dust is prevalent and typically active below $-15\,°C$, while some biological particles, such as specific bacteria, can initiate freezing at temperatures as high as $-1.5\,°C$ (Després et al., 2012; Huang et al., 2021; Janine Fröhlich-Nowoisky et al., 2016; Schnell and Vali, 1976; Vali et al., 1976). Quantifying total INPs, as well as distinguishing their biological and mineral fractions, provides critical insight into INP sources and atmospheric abundances.

Although offline drop freezing assay techniques have been employed for decades, recent intercomparison studies (DeMott et al., 2017, 2018d, 2025a; Lacher et al., 2024; Wex et al., 2015) affirm their effectiveness for ambient INP sampling. These methods are particularly valuable because they often capture INP concentrations across nearly the full heterogeneous freezing temperature range. Their simplicity makes them well-suited for long-term and remote deployments, as filters or other sample types can be easily collected and later analyzed offline. Long-term, multi-year INP records are critical for improving the representation of INP sources and their temporal evolution in earth system models (Burrows et al., 2022). Schrod et al. (2020) presented long-term measurements of deposition and condensation mode INPs from six diverse climatic regions, including the Amazon, Caribbean, central Europe, and the Arctic. Their near-continuous 24-hour samples—analyzed at –20, –25, and –30 °C—spanned over two years in some locations and showed relatively consistent INP concentrations across sites, generally within one order of magnitude. Similarly, Wex et al. (2019) reported comparable INP levels across multiple Arctic coastal sites, though they observed strong seasonal variability spanning several orders of magnitude, largely driven by the presence or absence of snow and sea ice. Freitas et al. (2023) documented a four-year record of Arctic INPs in Svalbard, which peaked during summer in conjunction with increased fluorescent biological particles. Schneider et al. (2021) reported 14 months of INP data from a Finnish boreal forest, showing seasonal alignment with primary biological aerosol particles (PBAPs),

including pollen. Gratzl et al. (2025) further linked seasonal INP fluctuations in the European sub-Arctic to fungal spores, particularly Basidiomycota, over the course of a year.

As recent studies have shown, long-term INP monitoring is especially powerful when integrated with detailed aerosol properties—such as mass concentration, size distribution, chemical composition, and optical characteristics—routinely measured by global in situ monitoring networks. The U.S. Department of Energy's Atmospheric Radiation Measurement (ARM) user facility is particularly well-suited for this purpose, with fixed sites and extended-duration mobile deployments that span a range of environments from the Arctic to the midlatitudes and the southern hemisphere. While INP measurements have been conducted at various ARM sites in the past, they were primarily user-driven and not part of the baseline measurement suite. These efforts have provided critical insights, including INP closure studies that reveal discrepancies between observed and predicted INPs, highlighting the need for improved parameterizations that may be missing key INP types (Knopf et al., 2021).

Recently, ARM has begun implementing baseline INP measurements at select sites, with coverage growing both spatially and temporally. The most extensive record to date spans nearly five years at ARM's fixed observatory in Oklahoma, USA. This paper outlines the availability of the valuable datasets at ARM sites, describing the sampling and offline analysis methods, data quality assurance pipelines, and access for the broader scientific community. A key aim is to raise awareness of these resources beyond current ARM users and encourage broader utilization by both experimentalists and modelers.

## 2 Sample collection and processing

### 2.1 ARM sites with existing INP measurements

#### 2.1.1 Fixed sites

Locations where INP measurements have been conducted or are currently underway are shown in Figure 1, with corresponding start and end dates, and filter collection frequency, listed in Table 1. For more up-to-date information on ARM observatories, visit https://www.arm.gov/capabilities/observatories. Detailed information on INP sampling, including field logs and filter metadata, is available at https://www.arm.gov/capabilities/instruments/ins. Filter samples are currently collected on a routine basis approximately every 6 days at two of the three fixed atmospheric observatories: the Southern Great Plains Central Facility in Lamont, Oklahoma (SGP C1; 314 m AMSL, 36.607° N, 97.488° W) and the North Slope of Alaska Central Facility in Utqiaġvik, Alaska (NSA C1; 8 m AMSL, 71.323° N, 156.615° W). Routine filter collections began at SGP C1 in October 2020 and are ongoing indefinitely, making it the first site globally with nearly five years of continuous INP measurements. At NSA C1, filter collection commenced in June 2025 and is likewise planned as a long-term effort.

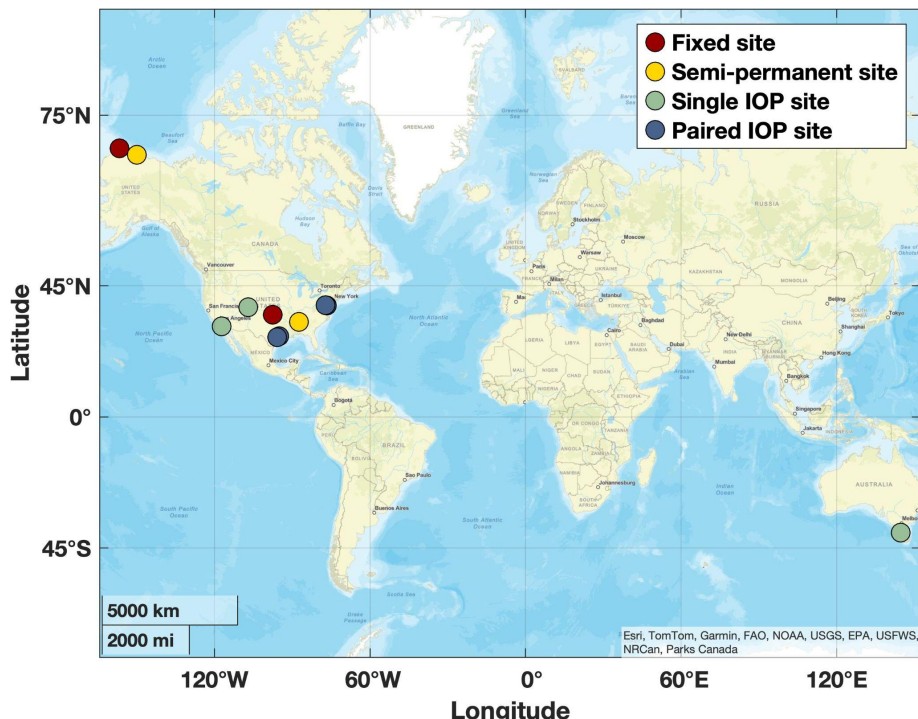

**Figure 1. Map of U.S. Department of Energy Atmospheric Radiation Measurement (DOE ARM) user facility sites where routine INP measurements have been established.** Red markers show fixed observatories, including Southern Great Plains (SGP C1) and North Slope of Alaska (NSA C1). ARM Mobile Facility (AMF) deployments are shown by yellow markers, while green and blue markers show IOP AMF deployment locations with single and paired sites, respectively. Paired sites indicate IOPs where main and supplemental site locations had simultaneous sample collections. Fixed and semi-permanent sites have single sample collection locations. See Table 1 for site details. Map was generated using Matlab with data from the Environmental Systems Research Institute.

An Intensive Observational Period (IOP) campaign, AGINSGP (Agricultural Ice Nuclei at SGP; Burrows, 2023), was conducted from September 2021 to May 2022. The objective of this deployment was to collect observations to better understand the drivers of variability in INP concentrations at the SGP locale, which are hypothesized to be influenced in part by regional emissions from fertile, organic-rich agricultural soils. Scientific users can submit requests to ARM to implement enhanced sampling strategies—such as increased temporal resolution, additional sampling sites, or entirely new locations— similar to the approach used during AGINSGP. Throughout the campaign, INP filters were collected approximately daily to support case study analyses following the field observations.

**Table 1. List of DOE ARM sites with INP measurements.** Also included are start and end dates and collection frequency of INP filters. Sites are indicated as either fixed, AMF, or ARM user-requested IOP (Intensive Observing Period). Sites that are continuous are labeled as such in the "INP filter end" column and those with "tbd" indicate an end date has yet to be determined.

| Site name | Site type | Site ID | INP filter start | INP filter end | Filter collection frequency |
|---|---|---|---|---|---|
| | | | | | |

| | | | | | |
|---|---|---|---|---|---|
| Southern Great Plains (SGP) Central Facility | fixed | SGP C1 | Oct 2020 | continuous | every 6 days |
| North Slope of Alaska (NSA) Central Facility | fixed | NSA C1 | Jun 2025 | continuous | every 6 days |
| Agricultural Ice Nuclei at SGP (AGINSGP) | IOP | SGP C1 | Apr 2022 | Apr 2022 | daily |
| Oliktok Point (OLI) Main Site | AMF | OLI M1 | Aug 2020 | Jun 2021 | every 6 days |
| Bankhead National Forest (BNF) Main Site | AMF | BNF M1 | Oct 2024 | tbd | every 6 days |
| Surface Atmosphere Integrated Field Laboratory (SAIL) Main Site | AMF | GUC M1 | Sep 2021 | Oct 2021 | every 6 days |
| Surface Atmosphere Integrated Field Laboratory (SAIL) second Supplemental Facility | AMF | GUC S2 | Nov 2021 | Jun 2023 | every 6 days |
| TRacking Aerosol Convection interactions ExpeRiment (TRACER) Main Site | AMF | HOU M1 | Jun 2022 | Sep 2022 | daily |
| TRacking Aerosol Convection interactions ExpeRiment (TRACER) third Supplemental Facility | AMF | HOU S3 | Jun 2022 | Sep 2022 | daily |
| Eastern Pacific Cloud Aerosol Precipitation Experiment (EPCAPE) Main Site | AMF | EPC M1 | Feb 2023 | Feb 2024 | every 6 days |
| Cloud And Precipitation Experiment at kennaook (CAPE-k) third Supplemental Facility | AMF | KCG S3 | Feb 2023 | Oct 2025 | every 6 days* |
| Coast-Urban-Rural Atmospheric Gradient Experiment (CoURAGE) Main Site | AMF | CRG M1 | Dec 2024 | Nov 2025 | every 6 days |
| Coast-Urban-Rural Atmospheric Gradient Experiment (CoURAGE) second Supplemental Facility | AMF | CRG S2 | Dec 2024 | Nov 2025 | every 6 days |
| CAPE-K-AEROSOLS | IOP | KCG S3 | Feb 2025 | Apr 2025 | daily |

*Filter durations vary due to the INS filter system operating only during clean sector "baseline" sampling periods. As such conditions were not always observed daily, individual 24-hour filter collections typically occur every ~6 days but may be more or less than 6 days depending on site-specific conditions.

## 2.1.2 Mobile facility sites

Scientific users can propose field campaigns (https://www.arm.gov/research/campaign-proposal) to deploy one of ARM's three Mobile Facilities (AMFs) in undersampled regions around the world. These mobile platforms provide comprehensive atmospheric measurements, including INP filter sampling. Deployments for the first and second mobile facilities (AMF1 and AMF2, respectively) typically span 6–18 months, with the third mobile facility (AMF3) being deployed for up to 5–8 years. Information on ARM INP measurements made at the AMFs is also included in Figure 1 and Table 1.

The first INP filters were collected as a part of the AMF3 at the Main Site in Oliktok Point, Alaska (OLI M1; 2 m AMSL,

70.495° N, 149.886° W), from August 2020 to June 2021. AMF3 was then relocated to the southeastern United States, where

filter collections began in October 2024 at the Main Site in Bankhead National Forest, Alabama (BNF M1; 293 m AMSL,

34.342° N, 87.338° W), and are ongoing.

INP filters were collected as a part of the AMF2 during the Surface Atmosphere Integrated Field Laboratory (SAIL; Feldman

et al., 2023) campaign in Crested Butte, Colorado. Sampling began at the Main Site (GUC M1; 2886 m AMSL, 38.956° N,

106.988° W) in September 2021 and continued through October 2021, before transitioning to the second Supplemental Facility

on Mt. Crested Butte (GUC S2; 3137 m AMSL, 38.898° N, 106.94° W), where collections continued for the duration of the

campaign from November 2021 to June 2023. AMF2 was subsequently deployed to Australia, where INP filters were collected

at the third Supplemental Facility during the Cloud And Precipitation Experiment at kennaook (CAPE-k) campaign, located

at the kennaook/Cape Grim Baseline Air Pollution Station on the northwestern tip of Tasmania (KCG S3; 67 m AMSL,

40.683° S, 144.690° E). This deployment began in February 2023 and concluded in October 2025. These samples were

collected during clean sector or "baseline" conditions—when winds originated from the southwest, transporting air masses

across the Southern Ocean that were free from local point source contamination. However, select samples were also captured

air masses from over Tasmania to help characterize potential local influences. Baseline information indicating when sector-

based sampling was active is available through the ARM Data Discovery portal

(https://adc.arm.gov/discovery/#/results/instrument_code::baseline).

The first INP filters collected using AMF1 were obtained in Texas during the TRacking Aerosol Convection interactions

ExpeRiment (TRACER) campaign (Jensen et al., 2023). Filters were collected at both the Main and third Supplemental Facility

sites in Houston (HOU M1: 8 m AMSL, 29.670° N, 95.059° W; HOU S3: 20 m AMSL, 29.328° N, 95.741° W) from June to

September 2022. The M1 site represented an urban environment, while the S3 site was rural. Due to the short duration of this

deployment, filters were collected approximately daily at both locations. Following TRACER, AMF1 was deployed to La

Jolla, California, as part of the Eastern Pacific Cloud Aerosol Precipitation Experiment (EPCAPE; Russell et al., 2024), where

INP filters were collected at the Main Site (EPC M1; 7 m AMSL, 32.867° N, 117.257° W) from February 2023 to February

2024. AMF1 resided in Maryland for the Coast-Urban-Rural Atmospheric Gradient Experiment (CoURAGE), where filter

collection occurred at both the Main and second Supplemental Facility sites in the Baltimore region (CRG M1: 45 m AMSL,

39.317° N, 76.586° W; CRG S2: 158 m AMSL, 39.422° N, 77.21° W). This deployment began in December 2024 and

continued through November 2025. As with TRACER, the M1 and S2 sites represent urban and rural environments,

respectively.

A very recent IOP campaign, known as CAPE-K-AEROSOLS (CAPE-k Summertime Single-Particle and INP Campaign),

was conducted from February to April 2025. This campaign aimed to improve understanding and predictability of Southern

Ocean aerosol concentrations, chemical composition, and sources, as well as their relationships to CCN and INPs. During this period, INP filters were collected approximately daily.

### 2.1.3 Tethered balloon system (TBS) deployments

ARM operates three TBSs, each capable of carrying payloads up to 50 kg on repeated vertical profiles through the atmospheric boundary layer, reaching elevations of approximately 1500 m AMSL depending on meteorological conditions and regulatory constraints. Detailed descriptions of the TBS systems are provided in Dexheimer et al. (2024). Vertically resolved INP filters have been collected during several ARM TBS deployments through ARM field campaign requests, using a customized miniaturized sampler. The TBS INP sampler design, filter preparations, deployments, and available data are described in detail in Creamean et al. (2025) and are only briefly mentioned here.

### 2.2 Filter preparation and sample collection

### 2.2.1 Fixed and AMF locations

Filter units are prepared following the methodology outlined in Creamean et al. (2024), with a brief summary provided here. Single-use Nalgene™ Sterile Analytical Filter Units are modified by replacing the original cellulose nitrate filters with 0.2-μm polycarbonate filters, backed by either 10-μm polycarbonate filters (both 47 mm diameter Whatman® Nuclepore™ Track-Etched Membranes) or 1-μm cellulose nitrate filters (47 mm diameter Whatman® non-sterile cellulose nitrate membranes), depending on the anticipated aerosol loading at each site. All components are pre-cleaned in-house following the procedure described in Barry et al. (2021). Filter units are disassembled and reassembled under ultraclean conditions inside a laminar flow cabinet with near-zero ambient particle concentrations, then sealed and stored individually in clean airtight bags until deployment.

Each sampling setup consists of the sterile, single-use filter units prepared at CSU, a totalizing mass flow meter (TSI Mass Flow Meter 5200-1 or 5300-1, TSI Inc.), a vacuum pump (Oil-less Piston Compressor/Vacuum Pump, Thomas), connecting tubing, and precipitation shields (Figure 2). Two identical filter assemblies operate in parallel: one collects primary filters for INP analysis, while the other collects duplicate filters, which serve either as backups or as archival samples available for user-requested analysis. The filter units are mounted open-faced and secured to the exterior of the AMF or other fixed-site infrastructure, protected from precipitation by shield covers. Each unit is connected via vacuum line tubing to the flow meter and vacuum pump, which are housed either within the main container or in an external pump enclosure, depending on the available space and site-specific conditions.

Upon completion of sampling (typically after 24 hours) the 0.2-μm filters containing the collected aerosol particles are carefully removed from the single-use filter units and stored frozen at approximately –20 °C in individual sterile Petri dishes (Pall®). As detailed in Table 1, the 24-hour samples are collected either daily or roughly every 6 days, depending on the goals

75  and duration of the measurement campaign. These samples are preserved on site until they can be transported in frozen batches
76  to CSU, where they remain frozen until they are processed and analysed.

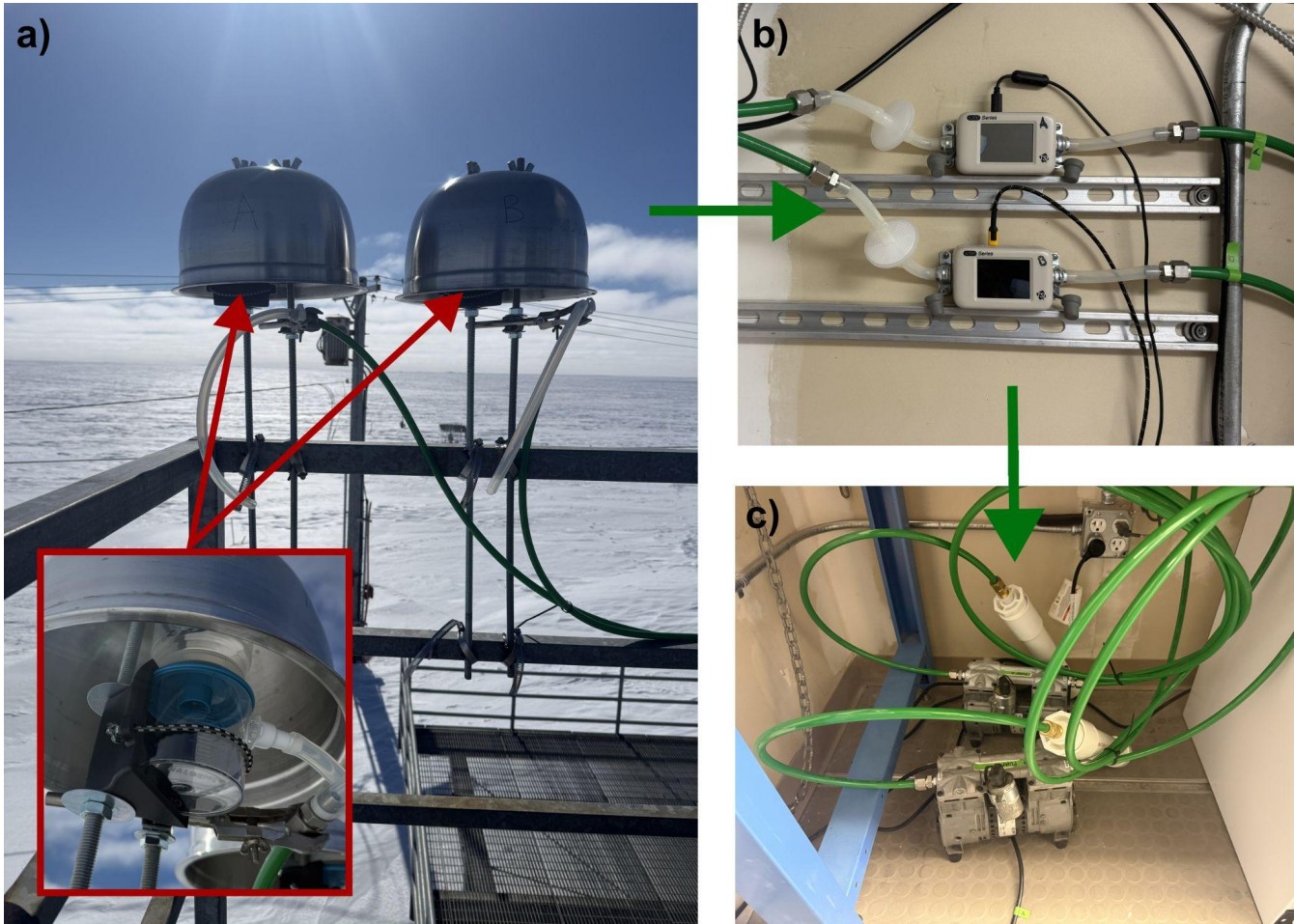

**Figure 2: Filter unit sampling apparatuses, including a) single-use, open-face filter units under precipitation shields which are connected via tubing to b) the mass flow meters and to c) the vacuum pumps.** Flow meters and pumps are always shielded from outside conditions. The inset in a) shows a magnified photo of a filter unit in a custom, 3D-printed filter holder. All photos are from the NSA C1 site.

## 2.3 Sample processing with the Ice Nucleation Spectrometer (INS)

The INS mimics immersion freezing of cloud ice through ambient aerosols serving as INPs by way of heterogeneous ice
nucleation. This technique provides quantitative information on the population of ambient aerosols that can facilitate cloud ice
formation at a wide range of subzero temperatures and, hence, INP concentration (e.g., 6 orders of magnitude). The INS (also
known as the Colorado State University (CSU) Ice Spectrometer) is supported with well-established experimental protocols
and has been applied in many diverse scenarios (e.g., Beall et al., 2017; DeMott et al., 2017; Hill et al., 2016; Hiranuma et al.,
2015; McCluskey et al., 2017, 2018; Suski et al., 2018). It is an offline analytical instrument used to quantify freezing

temperature spectra of immersion mode INP number concentrations from collected filter samples (Creamean et al., 2024). Each INS unit consists of two 96-well aluminum incubation blocks originally designed for polymerase chain reaction (PCR) plates, positioned end-to-end and thermally regulated by cold plates encasing the sides and base. Two INS instruments are operated side-by-side to increase sample processing throughput. The temperature measurement range of the INS is between 0 °C and approximately –27 to –30 °C.

For analysis, each filter is placed in a sterile 50 mL polypropylene tube with 7–10 mL of 0.1 μm-filtered deionized water, depending on expected aerosol loading. Lower volumes are used for cleaner environments to improve sensitivity. Samples are re-suspended by rotating the tubes end-over-end for 20 minutes. Dilution series are prepared using the suspensions and 0.1 μm-filtered deionized water, typically including 11-fold dilutions. Each suspension and its dilutions are dispensed into blocks of 32 aliquots (50 μL each) in single-use 96-well PCR trays (Optimum Ultra), alongside a 32-well negative control of filtered deionized water. The trays are placed in the aluminum blocks of the INS and cooled at 0.33 °C min$^{-1}$. Freezing is detected optically using a CCD camera with 1-second data resolution. HEPA-filtered $N_2$, pre-cooled near block temperature, continuously purges the headspace to prevent condensation build-up and warming of the aliquots.

The time between collection and analysis has ranged from one week to over a year. Beall et al. (2020) reported no significant differences in INP concentrations for samples stored between 1 and 166 days. In our internal quality checks, select samples stored for 1–2.5 years at $-20$ °C showed minimal differences (<1%), while larger differences (20–60%) were observed only in outlier cases associated with problematic original measurements.

**2.3.1 Heat and peroxide treatments**

Thermal and hydrogen peroxide ($H_2O_2$) treatments are used to probe INP composition, specifically targeting biologically-derived materials (Maki et al., 1974). Heat treatment involves heating 2.5 mL of sample suspension to 95 °C for 20 minutes to denature heat-labile INPs, such as proteins (Barry et al., 2023b, a; Hill et al., 2016, 2023; McCluskey et al., 2018b, c, a; Moore et al., 2025; Suski et al., 2018; Testa et al., 2021). Peroxide digestion is performed on a separate 2 mL aliquot by adding 1 mL of 30% $H_2O_2$ (Sigma-Aldrich®) to deionized water to create a 10% solution, followed by heating to 95 °C for 20 minutes under UVB illumination to generate hydroxyl radicals. Residual $H_2O_2$ is then neutralized using catalase (MP Biomedicals™, bovine liver). This process removes bio-organic INPs, as detailed in McCluskey et al. (2018c), Suski et al. (2018), and Testa et al. (2021). $H_2O_2$ has long been used as an oxidizing agent for degrading organic matter, originating in soil science nearly a century ago and later adopted across disciplines for removing organic material prior to chemical or physical analyses (McLean, 1931; Mikutta et al., 2005; Robinson, 1922; Schultz et al., 1999; Sequi and Aringhieri, 1977). In the presence of UV light, $H_2O_2$ photolyzes to form highly reactive hydroxyl radicals that can oxidize and structurally modify organic macromolecules, diminishing or eliminating their ice-nucleating activity (DeMott et al., 2023; Gute and Abbatt, 2020). Within the INP community, $H_2O_2$ treatments typically range from 10 to 35% (Beall et al., 2022; O'Sullivan et al., 2014; Perkins et al., 2020;

Roy et al., 2021; Teska et al., 2024; Tobo et al., 2019). We conducted recent tests showing minimal differences in ice-nucleating activity between 10% and 20% treatments; however, further validation is needed, and future community efforts should aim to establish a standardized protocol and concentration to ensure methodological consistency across studies.

The differences in freezing spectra before and after each treatment provide insights into INP composition—yielding total, heat-labile (biological), bio-organic, and inorganic (often mineral) INP concentrations. However, it is important to note that wet heating may lead to a slight decrease in ice nucleation activity in select mineral types (Daily et al., 2022). Blanks are included during peroxide digestion to monitor potential contamination from reagents. Treatments are typically applied to one-third of samples from each location. Due to the ongoing nature of this program, the numbers of treatments conducted on samples from any given site evolve continuously on a weekly basis. Exact sample sites, dates, identifiers, and other metadata regarding which samples undergo treatments can be accessed in real time via the publicly available field log on the INS website. In the data files available on the ARM Data Discovery portal, treatments are indicated by a flag: 0 for base/untreated data, 1 for heat-treated data, and 2 for peroxide-treated data.

## 3 From raw data to final product: processing and quality control

### 3.1 INP concentration and uncertainty calculations

INP concentrations are calculated at each temperature interval using the fraction of frozen droplets and the known total volume of air filtered, following Equation (1) (Vali, 1971):

$$K(\theta)\,(L^{-1}) \quad -\frac{\ln(1-f)}{V_{drop}} \times \frac{V_{suspension}}{V_{air}} \tag{1}$$

where $f$ is the proportion of frozen droplets, $V_{drop}$ is the volume of each droplet, $V_{suspension}$ is the volume of the suspension, and $V_{air}$ is the volume of air sampled (liters at standard temperature and pressure (STP) of 0 °C and 101.32 kPa). Specifically, the $V_{suspension}$ is the volume of 0.1 μm-filtered deionized water used to resuspend the particles from the filter (7–10 mL). The primary output of the INS is the freezing temperature spectrum of cumulative immersion mode INP number concentration, $K(\theta)$, from aerosols re-suspended from individual filters. INS output includes freezing temperature (°C), INP number concentration (L$^{-1}$ STP), 95% confidence intervals, and a treatment flag. Binomial confidence intervals are calculated following Agresti and Coull (1998), varying with the proportion of wells frozen. For example, freezing in 1 of 32 wells yields a confidence interval range of ~ approximately 0.2–5.0 times the estimated concentration, while 16 of 32 yields approximately 0.7–1.3 times the estimated concentration. The treatment flag denotes whether the suspension was base/untreated (total INPs; a flag of 0), heat-treated (biological INPs deactivated; a flag of 1), or H$_2$O$_2$-treated (organic INPs removed; a flag of 2). These values are derived from preliminary data files that include the processing date and time, freezing temperatures, and number of wells frozen (typically out of 32, each containing a 50 μL aliquot) per 0.5 °C interval.

Beyond Agresti and Coull confidence intervals, additional systematic and volumetric uncertainties associated with the INP measurements include instrumental and procedural sources that contribute to the overall error budget. The flow meter accuracy is ±4%, based on the TSI 5200 Series Gas Flow Meter Operation and Service Manual. Type T thermocouples have an estimated absolute accuracy of ±0.5 °C, though the relative uncertainty between data points is closer to ±0.1 °C (Perkins et al., 2020), and no measurable block temperature gradients have been observed in prior laboratory tests. Uncertainties in droplet and suspension volumes arise from pipetting and dispensing variability: ±1.3% for the larger pipette used in dilutions, ±2.5% for the smaller pipette, ±1.8% for the multipipetter, and ±0.1–0.8% for syringe-dispensed suspension volumes determined via gravimetric testing at CSU. Additionally, edge cases in freezing data (0/32 and 32/32) are not reported, and limits of detection (LOD) vary by sample based on total air and suspension volumes, with values below detection reported as −9999.

### 3.2 Quality control and assessment

To ensure the reliability and robustness of immersion freezing data from the INS, we implement a comprehensive quality control and assessment pipeline (Figure 3). This includes field sampling protocols, lab procedures, data validation, and instrument maintenance.

### 3.2.1 Field sampling quality control

Filter samples collected for offline INS processing are carefully monitored during field deployment. At both the start and end of each sampling period, the in-line pressure (kPa) and flow rate (standard liters per minute; Slpm) are recorded. These values are evaluated for anomalies such as significant pressure or flow changes, which may indicate issues like leaks in the filter unit, tubing, or system connections. To ensure accurate total air volumes are recorded, a totalizing mass flow meter logs flow every second during sample collection. This meter is annually sent to the manufacturer for recalibration. Units that deviate by more than 5% are returned to the manufacturer for servicing and recalibration. Field blanks are also prepared by briefly exposing unused filters to ambient air at the sampling site. All sites undergo field blank collection approximately once per month, with the exception of OLI M1, which is detailed in a Data Quality Report (DQR) that can be viewed under the "Description" for this site's INP data on the ARM Data Discovery Portal. Moving forward, all existing and new sites in the program will include routine monthly field blanks.

Routine maintenance for the field filter sampling system includes: (1) checking in-line temperature, pressure, and flow rate at the beginning and end of each sampling period, (2) inspecting precipitation shields and cleaning them as necessary, (3) ensuring single-use filter units are leak-free before deployment, (4) examining tubing and connection points for blockages or leaks, (5) verifying the performance of the vacuum pumps, which should sustain a 0.5 kPa vacuum, and (6) annual recalibration of the flow meters.

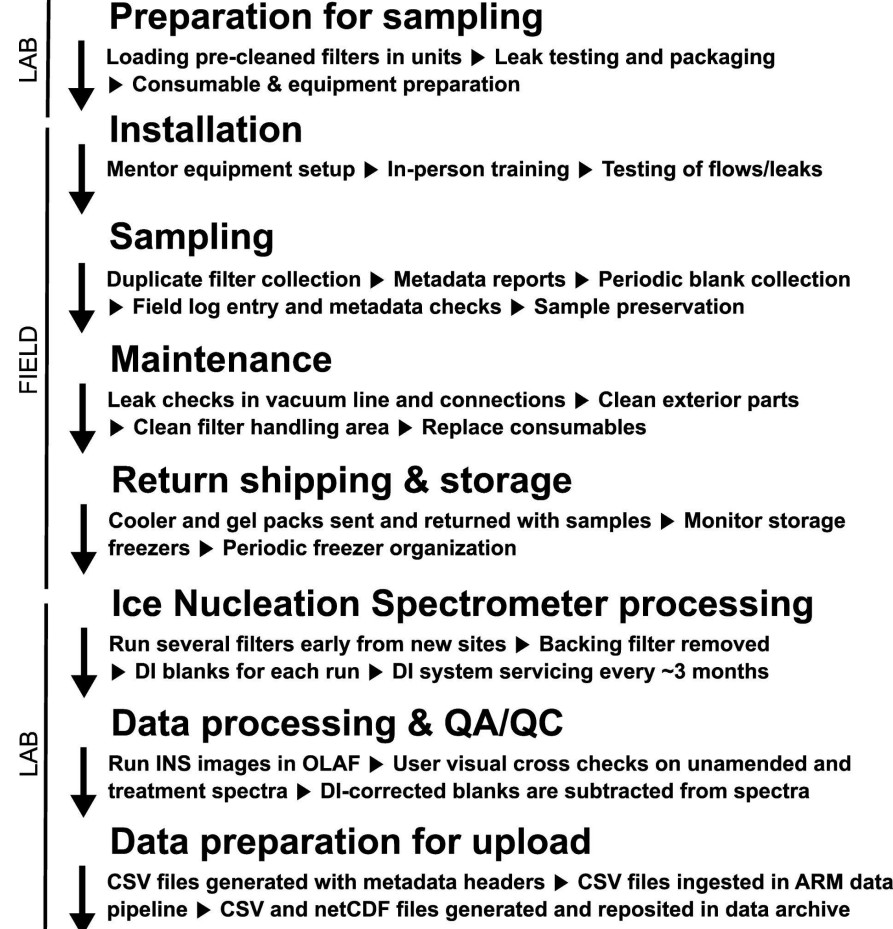

**LAB**

## Preparation for sampling
Loading pre-cleaned filters in units ▶ Leak testing and packaging ▶ Consumable & equipment preparation

## Installation
Mentor equipment setup ▶ In-person training ▶ Testing of flows/leaks

## Sampling
Duplicate filter collection ▶ Metadata reports ▶ Periodic blank collection ▶ Field log entry and metadata checks ▶ Sample preservation

**FIELD**

## Maintenance
Leak checks in vacuum line and connections ▶ Clean exterior parts ▶ Clean filter handling area ▶ Replace consumables

## Return shipping & storage
Cooler and gel packs sent and returned with samples ▶ Monitor storage freezers ▶ Periodic freezer organization

## Ice Nucleation Spectrometer processing
Run several filters early from new sites ▶ Backing filter removed ▶ DI blanks for each run ▶ DI system servicing every ~3 months

**LAB**

## Data processing & QA/QC
Run INS images in OLAF ▶ User visual cross checks on unamended and treatment spectra ▶ DI-corrected blanks are subtracted from spectra

## Data preparation for upload
CSV files generated with metadata headers ▶ CSV files ingested in ARM data pipeline ▶ CSV and netCDF files generated and reposited in data archive

**Figure 3: Flow diagram of quality assurance / quality control (QA/QC) protocols designed for DOE ARM INP data.** Quality assurance ensures that data meet established standards for both ARM management and scientific end users, while quality control involves systematic inspection and testing to verify that performance characteristics align with predefined specifications. DI = deionized.

### 3.2.2 Laboratory protocols

To minimize contamination from lab surfaces or consumables (e.g., pipet tips, PCR plates, tubes), we follow a stringent sample preparation protocol (Barry et al., 2021). Pipets are calibrated annually, and a 0.1-µm filtered deionized water blank is included with each INS run to correct for background INPs introduced during re-suspension or by the trays themselves. For peroxide digestion experiments, blanks with deionized water are included to detect potential contamination from $H_2O_2$ or catalase reagents. These are prepared using the same procedures as the actual samples to assess background INP levels and serve as a quality control check to determine whether reprocessing is necessary.

### 3.2.3 Instrument quality control and calibration

INS temperature accuracy is critical and maintained within ±0.2 °C, accounting for thermocouple uncertainty and ensuring no block temperature gradients develop over time. Each PCR block contains one thermocouple inserted just below the wells, and for each pair of blocks, the thermocouple readings are averaged. HEPA-filtered $N_2$ used to purge the PCR tray headspace is pre-cooled to prevent condensation build-up on plexiglass lids and warming the 50 µL aliquots during measurement. Camera images are captured every 20 seconds (approximately every 0.1 °C) during analysis to verify automated freezing detection. Each INS run is manually cross-checked against the recorded images to ensure proper identification of frozen wells. The deionized water blanks run with every sample serve a dual purpose: they act as a positive control as well and help monitor instrument drift, potential contamination, and the proper functioning of INS components (e.g., thermocouples, cameras). Routine lab maintenance of the INS includes: (1) cleaning plexiglass lids biweekly with Windex and deionized water, (2) monthly deep cleaning of the lab space, (3) monitoring copper piping for leaks of SYLTHERM™ XLT heat transfer fluid, and (4) watching the nitrogen tank depletion rate to detect leaks. We have confirmed the repeatability and reliability of the INS technique through replicate filter testing and campaign comparisons. Additionally, replicate filters have been analyzed to ensure comparability (Creamean et al., 2024).

### 3.3 Automated data processing algorithm

Historically, data produced by the INS have been analyzed manually using Microsoft Excel. In 2024, a data scientist was hired to develop the Open-source Library for Automating Freezing Data acQuisition from Ice Nucleation Spectrometer (OLAF DaQ INS), which now has its Version 1 completed. More information and the software itself are available at: https://github.com/SiGran/OLAF. Briefly, the OLAF DaQ INS software provides a graphical user interface that allows users to manually cross-check camera images taken during each INS run against the recorded well freezing data. Once image verification is complete, the program generates a CSV file with freezing data at every 0.5 °C interval, including the first instance of observed freezing to the nearest 0.1 °C. PCR wells containing deionized water are automatically subtracted from the sample wells for both the neat and serially diluted suspensions. These deionized water-corrected well freezing data are then converted to INP concentrations (per liter of air at STP) at each temperature bin using Equation (1). Binomial confidence intervals are calculated following Agresti and Coull (1998) and also converted to INP $L^{-1}$ using the same equation. For each temperature bin, the program selects the INP concentration from the least dilute sample that remains statistically valid. When a dilution reaches its statistically significant limit, the program moves to the next most dilute sample.

In cases where INP concentrations decrease with decreasing temperature (an artifact sometimes introduced by the stochastic nature of serial dilution measurements) the program automatically adjusts the values to enforce monotonicity. We are currently developing an additional QC flag for OLAF-generated data files to indicate which data points were adjusted due to monotonicity-related corrections. This correction was not applied prior to OLAF when files were generated manually. Because blank subtraction can also produce this artifact, the correction is applied after the blank subtraction step. Specifically, if a

blank-corrected value falls below the lower 95% confidence bound of the uncorrected value, the program replaces it with the previous bin's value and propagates the upper confidence interval using the root mean square of the current and previous intervals. This correction is applied only when the number of affected bins remains below a user-defined threshold (10% of total temperature bins per sample); if exceeded, those bins are flagged with an error signal ($-9999$). At ground-based terrestrial sites, corrections are almost entirely due to dilution stochasticity and rarely result from blank subtraction, whereas marine or other low-aerosol loading environments tend to experience a higher frequency of corrections related to blank subtraction. Finally, the software compiles all blank-corrected data across treatments (base/untreated, heat, and peroxide) into a single output file, including treatment flags for each sample.

### 3.4 Ingesting processed INP data into ARM Data Discovery

The final step in making INS-derived INP data publicly available is ingestion into the ARM Data Discovery portal. This begins with the CSV files generated during INS processing, which are passed through an automated pipeline that standardizes them into a universal format used across all ARM datasets. This format includes all necessary metadata headers and timestamps. During ingestion, the ARM Data Quality Office (DQO) evaluates the data by reviewing plots and statistical metrics of the INP data. If any issues are identified, the DQO works with the mentor team to resolve them. This dual-level review, by both scientific mentors and the DQO, ensures the robustness and reliability of the final data products. Once approved, the data are published at the "a1" level, which denotes that calibration factors have been applied, values have been converted to geophysical units, and the dataset is considered final. These files are available in NetCDF and/or ASCII-CSV formats and can be accessed by placing a data order through the ARM Data Discovery portal. A free ARM account is required to request and download the data.

## 4 Applications of ARM INP data

### 4.1 Temporal trends in INP concentrations from long-term monitoring

As the first established fixed site, SGP C1 hosts nearly five consecutive years of INP concentration data (Figure 4). Untreated (i.e., base, or total INP) measurements, collected approximately every six days, are publicly available. Long-term datasets such as this are invaluable for examining the annual cycle of INPs in detail. For instance, Figure 4 reveals a pronounced seasonal pattern, with INP concentrations peaking during the fall/winter months (October–January), particularly at warmer freezing temperatures (e.g., $> -10\,°C$). At colder temperatures (e.g., $\leq -15\,°C$), the seasonal cycle is less distinct. Although INPs active at the warmest temperatures ($\geq -6\,°C$) were relatively rare, the few observed events tended to coincide with the fall/winter peak. This site is influenced by surrounding agricultural activities, which may contribute to the observed seasonal variability in INPs; however, a comprehensive source attribution is beyond the scope of this manuscript. Our intent here is to highlight the completeness and continuity of the SGP dataset and its utility. These measurements support both observational studies of INP variability and source characterization, and model evaluation efforts such as Knopf et al. (2021).

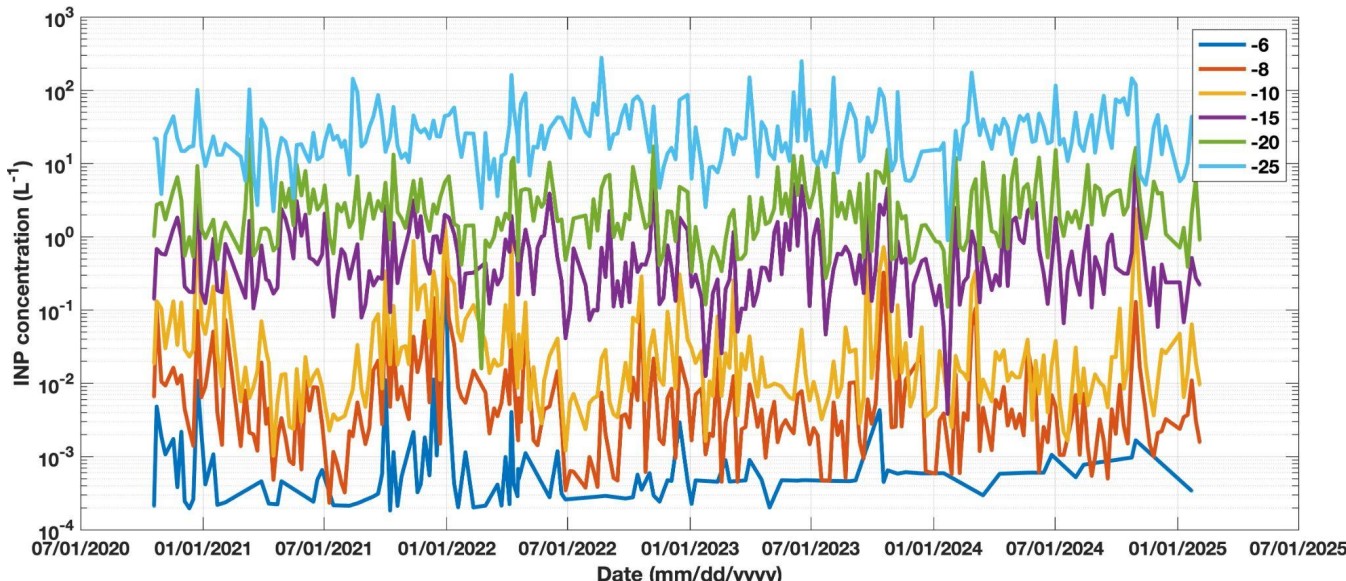

**Figure 4: Complete time series of INP concentrations at select temperatures from the SGP C1 site that are currently publicly available on DOE ARM Data Discovery.** Each line shows cumulative INP concentrations per liter of air ($L^{-1}$) at freezing temperatures designated in the legend (in °C). Data are from 247 total processed filter samples.

## 4.2 Characterizing INP types through heat and peroxide treatments

In addition to the time series of total INP concentrations, approximately one-third of the samples undergo specific heat and peroxide treatments to help identify broad classes of INP types. These treatments target: (1) heat-labile INPs, such as proteins commonly associated with biological particles; (2) heat-stable organics, isolated via hydrogen peroxide treatment; and (3) the remaining, largely inorganic fraction, which is often attributed to mineral dust (Barry et al., 2023a, 2025; Creamean et al., 2020; DeMott et al., 2025a; Hill et al., 2016; McCluskey et al., 2018c; Schiebel et al., 2016; Suski et al., 2018; Testa et al., 2021; Tobo et al., 2019). Figure 5 provides an example of the relative contributions of these INP types over time at SGP C1, shown as a percentage of total INPs at two temperatures. The fraction of "biological" INPs is derived by subtracting the heat-treated INP spectrum from the untreated spectrum. The "organic" component is isolated by subtracting the peroxide-treated spectrum from the heat-treated spectrum. The residual "inorganic" fraction is estimated by subtracting the peroxide-treated spectrum from the untreated spectrum.

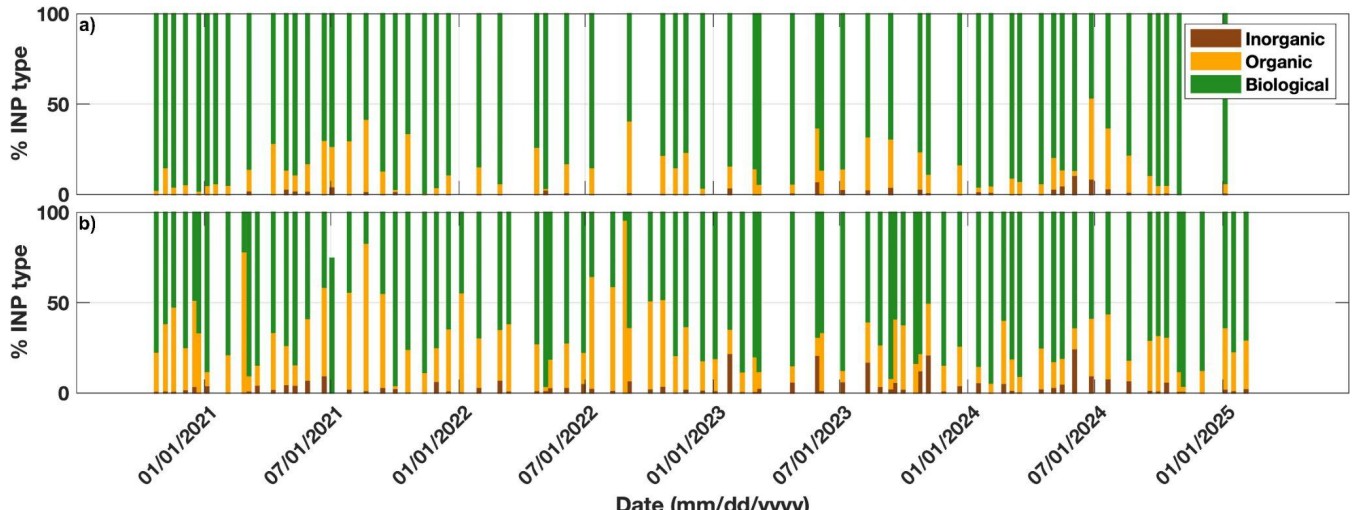

Figure 5: Relative abundance of INP type at the SGP C1 site at a) –15 °C and b) –25 °C that are currently available on DOE ARM Data Discovery. INP types are determined through heat and peroxide treatments. We assume that the reduction of INPs from heat are biological in nature (e.g., heat labile proteins) while the reduction of INPs from peroxide, UV, and heat are organic (e.g., heat labile organics). INPs remaining (unaffected) by both treatments are inorganic (e.g., mineral dust). Data are from 84 samples that have been heat- and peroxide-treated (34% of the processed SGP samples in Figure 4).

These unique long-term data offer insights into the seasonal variability and relative importance of different INP sources. For instance, at –15 °C, biological INPs dominate at SGP, with smaller contributions from organics and inorganics. The inorganic component becomes more apparent during the summer months, likely associated with dry, dusty conditions on agricultural lands (Evans, 2025; Ginoux et al., 2012). At –25 °C, the relative contributions of organic and inorganic INPs increase, yet biological INPs still remain the dominant type overall. Although the Great Plains region is periodically influenced by dust events, its agricultural soils are rich in biological material (Delgado-Baquerizo et al., 2018; Garcia et al., 2012; Hill et al., 2016; Kanji et al., 2017; O'Sullivan et al., 2014; Pereira et al., 2022; Steinke et al., 2016; Suski et al., 2018; Tobo et al., 2014), which distinguishes it from more arid, desert regions where mineral dust may dominate. These compositional insights are particularly valuable for users interested not only in INP abundance but also in potential sources. The treatment data can be used in combination with aerosol composition and meteorological observations at SGP C1 (and other ARM sites), and air mass trajectory analysis to further constrain the origins of INPs.

**4.3 Seasonal INP variability across sites**

INP data can be meaningfully compared across a diverse range of sites throughout the year, as illustrated in Figures 6 and 7 for –10 °C and –20 °C, respectively. The purpose of these figures is to highlight the diversity of INP concentrations across a range of environments and to demonstrate the value of consistent INP measurements at multiple sites. Each site shown represents a distinct setting: EPC M1 is a coastal marine site in California; GUC S2 is located at high elevation in the Colorado Rocky Mountains; the HOU sites include both urban and rural environments in Texas; OLI S3 is situated in a coastal oilfield region of northern Alaska within the Arctic; and SGP represents a high plains agricultural site in the central U.S. These are the

91 sites for which data are currently available through ARM Data Discovery, with additional datasets forthcoming for sites in

92 Tasmania, northern Alaska, and the northeastern and southeastern United States.

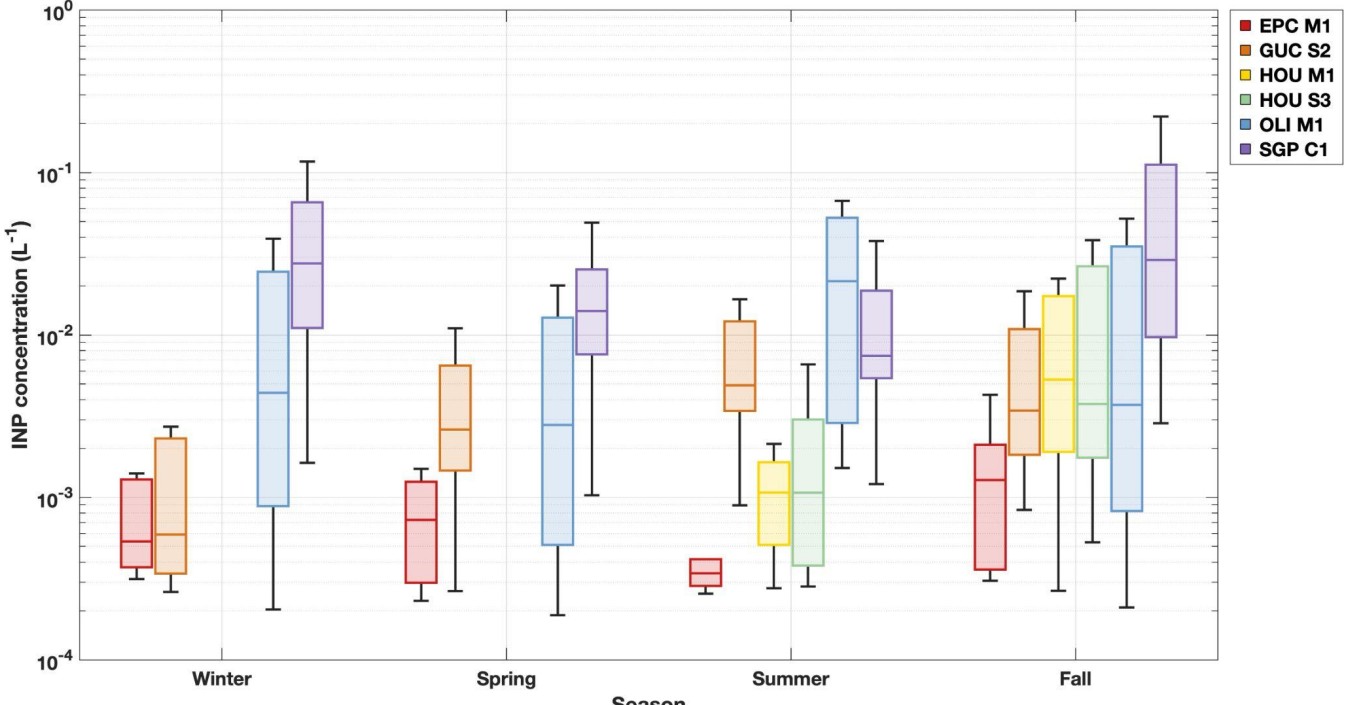

93

**Figure 6: Seasonal INP concentrations at –10 °C at all fixed and mobile facility sites currently available from the DOE ARM Data Discovery.** Data are presented in box-and-whisker format, with the middle line being the median (50th percentile), box edges representing the 25th and 75th percentiles, and the whiskers representing data within 1.5× the interquartile range. The numbers above each median line indicate the number of data points that went into each bar.

98 Several noteworthy patterns emerge from these intercomparisons. At –10 °C, where INPs are likely dominated by biological

99 materials (Huang et al., 2021; Kanji et al., 2017), many sites exhibit clear seasonal cycles, though the timing and magnitude

00 of these cycles differ. For instance, SGP shows elevated INP concentrations in the winter and fall, consistent with agricultural

01 activity and associated emissions during that time. In contrast, GUC exhibits higher concentrations in summer, which aligns

02 with the seasonal exposure of vegetation and the wintertime snow cover typical of the Colorado Rocky Mountains. Similarly,

03 the Arctic coastal site OLI displays peak concentrations in summer, even exceeding those at the midlatitude SGP site. This is

04 consistent with findings highlighting the biological productivity of Alaskan Arctic waters and tundra in May through

05 September leading to increased airborne INPs (Barry et al., 2025; Creamean et al., 2018a, 2019; Eufemio et al., 2023; Fountain

06 and Ohtake, 1985; Nieto-Caballero et al., 2025; Perring et al., 2023; Rogers et al., 2001; Wex et al., 2019), despite the presence

07 of extensive oil and gas infrastructure near OLI that impacts the aerosol composition (Creamean et al., 2018b; Gunsch et al.,

08 2017). However, a few important considerations should be noted. Field blanks were not collected at OLI; instead, a laboratory

09 blank was used to subtract background INPs. This approach may lead to artificially elevated concentrations, as lab blanks

typically have lower background levels than field blanks due to reduced handling and exposure. Additionally, the OLI data

represent a single summer season, whereas the SGP data span four summers. If the OLI summer was anomalous, this could

skew comparisons. These factors should be carefully considered when interpreting or using the OLI dataset.

Conversely, EPC recorded the lowest INP concentrations among the sites, likely due to its exposure to clean marine air masses,

which are generally associated with INP levels lower than terrestrial environments (e.g., DeMott et al., 2016; McCluskey et

al., 2018b; Welti et al., 2020). Interestingly, both the urban and rural sites in HOU exhibited similar INP concentrations during

the summer and fall, despite the common assumption that urban emissions are generally poor sources of INPs (Bi et al., 2019;

Cabrera-Segoviano et al., 2022; Chen et al., 2018; Hasenkopf et al., 2016; Ren et al., 2023; Schrod et al., 2020; Tobo et al.,

2020; Wagh et al., 2021; Yadav et al., 2019; Zhang et al., 2022; Zhao et al., 2019). The results from OLI and HOU collectively

suggest that nearby regional marine sources can substantially influence INP concentrations, even in regions characterized by

high levels of industrialization or urbanization.

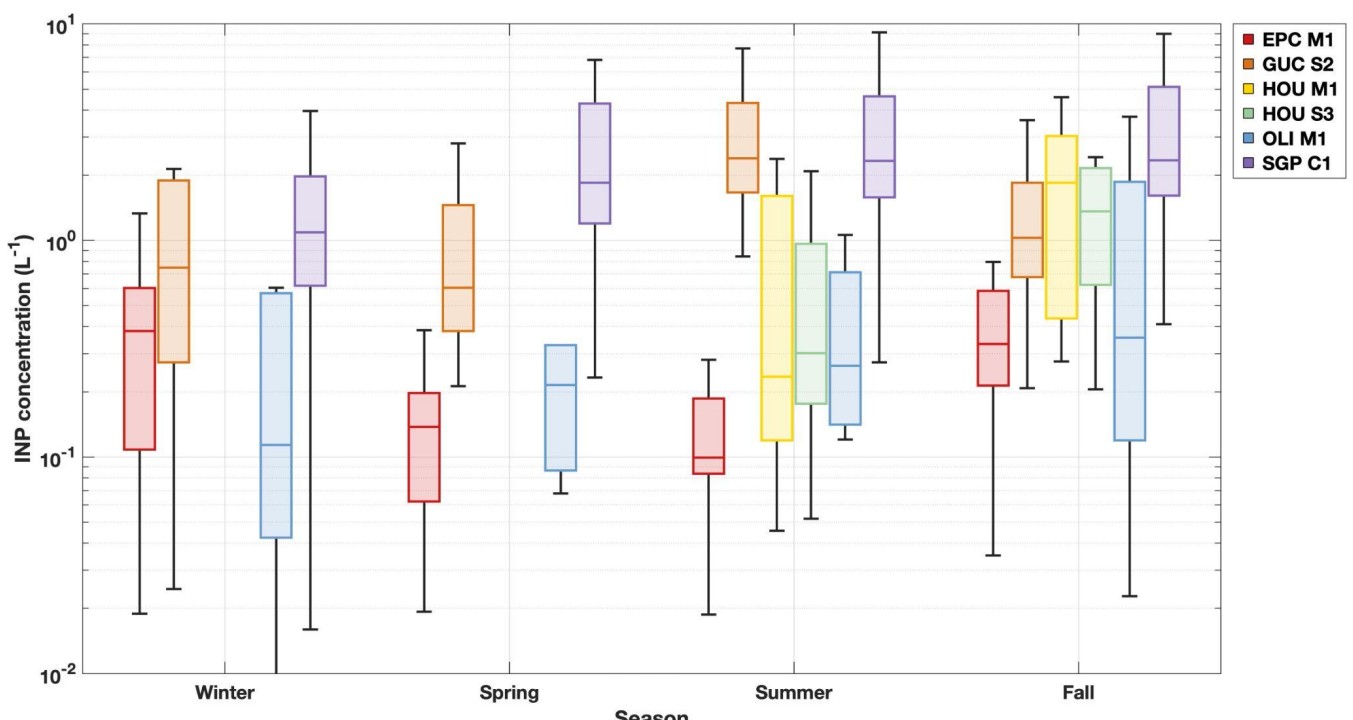

**Figure 7: Same as Figure 6, but for seasonal INP concentrations at −20 °C at all fixed and mobile facility sites currently available**
**from the DOE ARM Data Discovery.** Note the scale of the INP concentration axis is higher than Figure 6.

At −20 °C, seasonal patterns in INP concentrations remain evident across most sites, but notable differences emerge compared

to the −10 °C data. INP concentrations at the two HOU sites remain comparable, consistent with the pattern observed at warmer

temperatures. However, one of the most striking differences is that OLI, which had among the highest concentrations at −

10 °C, no longer stands out; instead, it shows significantly lower INP levels than SGP. This shift suggests that SGP may have a more prominent source of mineral dust or cold-temperature-active organic INPs than the Arctic coastal OLI site. This interpretation is consistent with known regional differences, as the U.S. midlatitudes, including the central plains where SGP is located, coexist with more prominent dust emissions compared to the North American Arctic (e.g., Ginoux et al., 2012; Rodriguez-Caballero et al., 2022; Song et al., 2021). Interestingly, INP concentrations at OLI are now more comparable to those at EPC, likely reflecting the marine influence at both locations, which generally has lower INP concentrations relative to continental sources.

These INP measurements are consistent with many principal investigator-led datasets collected at other ARM-supported locations, such as those that employ the Colorado State University Ice Spectrometer (see Table 3). The INS that is used to produce the ARM INP data is almost identical to the Ice Spectrometer. This opens opportunities for broader comparisons to campaigns such as the 2017–2018 MARCUS (Measurements of Aerosols, Radiation, and Clouds over the Southern Ocean; DeMott et al., 2018b; McCluskey et al., 2018c; McFarquhar et al., 2019, 2021; Niu et al., 2024; Raman et al., 2023) and 2016–2018 MICRE (Macquarie Island Cloud and Radiation Experiment; DeMott et al., 2018a; Marchand, 2020; McCluskey et al., 2023; Niu et al., 2024; Raman et al., 2023) campaigns in the Southern Ocean, 2019–2020 MOSAiC (Multidisciplinary drifting Observatory for the Study of Arctic Climate; Barry et al., 2025; Creamean et al., 2022; Shupe et al., 2021, 2022) campaign in the Arctic Ocean, 2019–2020 COMBLE (Cold-Air Outbreaks in the Marine Boundary Layer Experiment; DeMott and Hill, 2021; DeMott et al., 2025b; Geerts et al., 2021, 2022) campaign along the Norwegian Arctic coast, 2018–2019 CACTI (Cloud, Aerosol, and Complex Terrain Interactions; DeMott and Hill, 2020; Testa et al., 2021; Varble et al., 2019) campaign in agricultural regions of South America, the 2019 AEROICESTUDY (Aerosol-Ice Formation Closure Pilot Study; Knopf et al., 2020, 2021) and 2014 INCE (Ice Nuclei Characterization Experiment; DeMott et al., 2015) at SGP, and the 2015 ACAPEX (ARM Cloud Aerosol Precipitation Experiment; DeMott and Hill, 2016; Fan et al., 2014; Leung, 2016; Levin et al., 2019; Lin et al., 2022) study off the coast of California. These complementary datasets are also publicly available through ARM Data Discovery, but labeled as "icespec" (or "icespec-air" for aircraft measurements).

**Table 3. List of previous PI-led DOE ARM field campaigns with comparable INP data to the INS.** Includes measurement location, start and end dates, filter collection details, and DOI for the INP measurements. Data from earlier studies do not have available DOIs. Note all of these campaigns are AMF deployments. RV is abbreviated for Research Vessel.

| Field campaign name | Location | INP filter start | INP filter end | Filter collection details | DOI (https://doi.org/) |
|---|---|---|---|---|---|
| Measurements of Aerosols, Radiation, and Clouds over the Southern Ocean (MARCUS) | Southern Ocean on the *Aurora Australis* | Oct 2017 | Apr 2018 | continuous; 24- to 48-h samples | 10.5439/1638968 |
| Macquarie Island Cloud and | Macquarie Island, | Mar | Mar 2018 | continuous; 48- to | 10.5439/1638330 |

| | | | | | |
|---|---|---|---|---|---|
| Radiation Experiment (MICRE) | Australia | 2016 | | 72-h samples | |
| Multidisciplinary Drifting Observatory for the Study of Arctic Climate (MOSAiC) | Arctic Ocean on the *RV Polarstern* | Oct 2019 | Oct 2020 | continuous; 72-h samples | 10.5439/1804484 |
| Cold-Air Outbreaks in the Marine Boundary Layer Experiment (COMBLE) | Andenes, Norway | Dec 2019 | Mar 2020 | during CAOs; 6- to 74-h samples | 10.5439/1755091 |
| Cloud, Aerosol, and Complex Terrain Interactions (CACTI) | Villa Yacanto, Argentina | Oct 2018 | Apr 2019 | quasi-continuous; 8-h samples | 10.5439/1607786 |
| Cloud, Aerosol, and Complex Terrain Interactions (CACTI) | Sierras de Córdoba, Argentina | Nov 2018 | Dec 2018 | flight duration; various sample durations | 10.5439/1607793 |
| Aerosol-Ice Formation Closure Pilot Study (AEROICESTUDY) | SGP | Oct 2019 | Oct 2019 | continuous; 12- to 24-h samples | 10.5439/1637710 |
| Ice Nuclei Characterization Experiment (INCE) | SGP | Apr 2014 | Jun 2014 | continuous; 24-h samples | none |
| ARM Cloud Aerosol Precipitation Experiment (ACAPEX) | Pacific Ocean on the ARM G-1 aircraft | Jan 2015 | Mar 2015 | flight duration; 10-min to 3-h samples | none |

## 5 Community use and limitations of ARM INP data

We present a comprehensive dataset of immersion mode INP concentrations from multiple sites across the United States and beyond. Most of these data are publicly available through the DOE ARM Data Discovery portal (https://adc.arm.gov/discovery/). On the portal, data from fixed sites and AMF deployments can be found by searching for "INP," while data collected via ARM tethered balloon systems can be found by searching for "TBSINP." DOIs for INP and TBSINP are https://doi.org/10.5439/1770816 and https://doi.org/10.5439/2001041, respectively. For sites with ongoing measurements, data are routinely uploaded as batches of samples are processed using the INS. Upcoming INP datasets from the CAPE-k (KCG S3), CoURAGE (CRG M1 and S2), BNF (M1), and NSA (C1) sites will also be made available in the near future. These ARM-based INP measurements are directly comparable to other principal investigator-led datasets collected in previous studies at a wider range of locations, allowing for meaningful cross-site comparisons.

Importantly, duplicate filters are collected at most sites and preserved frozen for potential future analyses. Researchers interested in obtaining additional INP data on unprocessed samples or conducting their own supplementary aerosol physicochemical analyses can request these archived samples by submitting an ARM Small Campaign Request

(https://www.arm.gov/guidance/campaign-guidelines/small-campaigns) with the option to contact the ARM INP mentor team

(co-authors on this manuscript) with questions. At many of the sites listed in Table 1, only a subset of collected filters has been

processed to date. Therefore, users with specific dates or time periods of interest are encouraged to reach out to the mentor

team to request new analyses, including specialized treatments. A detailed filter collection log is available on the ARM INS

homepage (https://www.arm.gov/capabilities/instruments/ins) to help guide these inquiries. INP data from future campaigns

requested by researchers will also be made accessible to the broader research community.

The DOE ARM baseline INP measurements provide valuable long-term and IOP-based observations but have several

limitations that users should be aware of. First, these measurements do not account for time dependence in freezing behavior,

which is generally less significant than temperature dependence (Ervens and Feingold, 2013). Second, sampling assumes

collection of the total aerosol size distribution; however, this has not been explicitly tested, so the exact size range collected is

uncertain. The 0.2-μm filters we use have reduced transmission efficiency for particles around 150 nm (down to 65–78%), but

generally exhibit very high collection efficiency at most sizes (Spurny and Lodge, 1972). Third, because filters are collected

over 24 hours, typically every six days, short-term or episodic INP events may be missed, although higher-frequency sampling

can be requested. Fourth, these measurements are made at the surface and may not fully represent the INP population at cloud

level, though cloud-surface coupling analyses (e.g., Creamean et al., 2021; Griesche et al., 2021) and TBS INP data (Creamean

et al., 2025) can help bridge this gap. Lastly, not all samples are subjected to treatments unless requested, and, as noted by

Burrows et al. (2022), the absence of co-located baseline measurements of aerosol composition (particularly dust, sea spray,

and primary biological aerosol particles) limits the ability to fully constrain INP sources and improve parameterizations in

models.

## Data availability

INP and TBSINP data are available from the DOE ARM Data Discovery portal (https://adc.arm.gov/discovery/) under DOIs

https://doi.org/10.5439/1770816 (Creamean et al., 2024) and https://doi.org/10.5439/2001041 (Creamean et al., 2025),

respectively.

## Author contributions

JMC and AT conceptualised the INP mentor program. CCH and MV conducted the sample and data analysis for the INP data

that are publicly available for download from the DOE ARM Data Center. CCH and JMC were additionally responsible for

instrument installation and maintenance at the sites. All authors contributed to the writing of this manuscript.

## Competing interests

None of the authors has any competing interests

## Disclaimer

Publisher's note: Copernicus Publications remains neutral with regard to jurisdictional claims made in the text, published maps, institutional affiliations, or any other geographical representation in this paper. While Copernicus Publications makes every effort to include appropriate place names, the final responsibility lies with the authors.

## Acknowledgements

This work was supported by the Office of Biological and Environmental Research within the U.S. Department of Energy (DOE) through the Atmospheric Radiation Measurement (ARM) user facility. JMC, CCH, and MV received support under DOE contract no. DE-0F-60173. We gratefully acknowledge James Mather for his invaluable support in the development and implementation of the INP program. We also extend our sincere thanks to the ARM site staff for their significant assistance with instrument installation, sample collection, and logistics. We gratefully acknowledge Thomas C. J. Hill for his foundational role as co-mentor alongside JMC during the inception of this program, and for his enduring guidance and expertise. He is now enjoying a well-earned retirement in Australia. ChatGPT was used to assist in editing and improving the wording of this manuscript.

## Financial support

This research has been supported by Argonne National Laboratory for the DOE under contract DE-0F-60173.

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
