# Peer review of "Long-term measurements of ice nucleating particles at Atmospheric"

_Earth System Science Data, 2025_

## Author Response (AR1)

We thank the reviewers for their thoughtful feedback, which has helped us to further improve and strengthen the manuscript. Our detailed responses to the final reviewer comments are provided below in blue. All revisions are visible in the track-changes version of the revised manuscript included after the responses.

**Reviewer #1**

This manuscript describes the U.S. DOE ARM program's immersion-mode ice-nucleating particle (INP) dataset, spanning fixed observatories and ARM Mobile Facility (AMF) deployments, as well as vertically resolved sampling with tethered balloon systems (TBS). The paper documents sampling protocols, INS processing (including heat and $H_2O_2$ treatments), and automated analysis with the OLAF DaQ INS software, and points readers to data access via ARM Data Discovery. The work represents a highly valuable, standardized, multi-site record with strong potential for model evaluation, parameterization development, and process studies.

Specific comments A. Major issues

1. Sampling duration vs. frequency (Table 1; Section 2.2.1, lines 177–181): Table 1 reports ~6-day collection frequency at several sites, whereas Section 2.2.1 states that sampling is "typically after 24 hours". Please reconcile this apparent contradiction explicitly in the text and encode, per site and period, the actual filter exposure duration in the metadata. If some sites use multi-day exposures, discuss the potential for on-filter chemical aging (photo-/heterogeneous oxidation) to bias INP spectra—particularly depressing labile organics at warm T ($-5$ to $-15$ °C) and enriching refractory INPs. Consider a sensitivity check comparing daily versus 6-day exposures during IOPs.

This wording may have been unclear as originally written. We collect 24-hour filters approximately every 6 days, resulting in ~5-day gaps between collections. To clarify, we specified this in the Table 1 footer and added the following sentence to Section 2.2.1: "As detailed in Table 1, the 24-hour samples are collected either daily or roughly every 6 days, depending on the goals and duration of the measurement campaign." Because these are discrete 24-hour samples rather than multi-day exposures, they are not subject to potential issues of on-filter chemical aging.

2. Coordinate error (Section 2.1.1, lines 82–84; Table 1): The SGP site is given as 97.488° E, but it should be 97.488° W. Please correct this typo and ensure it propagates to Table 1, Figure 1, and metadata exports.

This was just a typo in Section 2.1.1. We have corrected it there. It was already the correct coordinate (in °W) in the map and in metadata exports.

3. Automated monotonicity correction (Section 3.3, lines 297–302; Fig. 3): Section 3.3 describes adjusting bins to enforce monotonicity when blank subtraction yields decreasing K(T). Later it says bins exceeding a threshold are "flagged with an error signal". Please quantify how often this correction is applied per sample/site, expose a counter and QC flag in the NetCDF, and—if possible—retain pre-correction values.

As noted in Section 3.3, bins that exceed the error threshold during blank subtraction are assigned an error value of −9999. We have revised the text in this section to clarify this assignment. Regarding the retention

of pre-corrected values, these are not included in the final data product to avoid potential confusion or inadvertent use of uncorrected data instead of the finalized QA/QC-processed values.

It is important to note that OLAF was developed and fully operational within the past year, and has only been applied to recent datasets, including CRG, BNF, and recent SGP samples; earlier datasets were processed manually. We already stated this in the manuscript at the beginning of Section 3.3 but included this statement in the newly-added text for clarity: "This correction was not applied prior to OLAF when files were generated manually."

We agree that implementing a monotonicity QC flag in the NetCDF files is a valuable improvement and will begin adding this feature to existing OLAF-generated files, as well as all future datasets. We added a statement to Section 3.3 delineating this: "We are currently developing an additional QC flag for OLAF-generated data files to indicate which data points were adjusted due to stochasticity-related corrections."

4. Uncertainty budget (Equation 1, Section 3.1, lines 232–239): Beyond Agresti–Coull intervals, please include systematic and volumetric terms: flow meter accuracy (±2%), temperature measurement (±0.2 °C, including block gradients), droplet volume tolerance, and suspension volume uncertainties. Discuss edge cases (0/32 and 32/32) and how LOD/LOQ are reported. Providing per-bin combined uncertainties will improve usability.

We appreciate the reviewer's suggestion to expand the uncertainty budget beyond the Agresti-Coull intervals. We have now included additional sources of systematic and volumetric uncertainty at the end of Section 3.1. Specifically:

- Flow meter accuracy: ±4%, based on the 5200 Series Gas Flow Multi-Meter Operation and Service Manual (TSI, US).
- Thermocouple accuracy: Type T thermocouples are estimated at ±0.5 °C, though relative temperature uncertainty between data points is closer to ±0.1 °C (Perkins et al., 2020).
- Block temperature gradients: No measurable gradient has been observed in previous lab tests. Uncertainty is sufficiently captured by thermocouple range.
- Droplet volume tolerance:
    - Larger pipette used for DI water in dilutions: ±1.3%
    - Smaller pipette used for "sample" in dilutions: ±2.5%
    - Multipipetter: ±1.8%
- Suspension volume uncertainty: Currently dispensed using 20 mL syringes; a gravimetric test was conducted in the CSU lab to quantify this uncertainty and ranges from ±0.1% to ±0.8 % depending on the dispensed volume.

We also added clarification on edge cases and detection limits:

- 0/32 and 32/32 freezing cases are not reported.
- The LOD varies by sample depending on the total air and suspension volumes. Values below detection are reported as −9999.

Perkins, R. J., Gillette, S. M., Hill, T. C. J., and DeMott, P. J.: The Labile Nature of Ice Nucleation by Arizona Test Dust, ACS Earth Space Chem., 4, 133–141, https://doi.org/10.1021/acsearthspacechem.9b00304, 2020.

5. Blank strategy and Oliktok (Sections 2.1.2–2.1.3, Table 1; Section 3.2.1, lines 254–255): Clarify, in a table and in the data files, which sites have field versus lab blanks. For Oliktok (OLI), state explicitly that only lab blanks were available and flag affected samples in the data (e.g., "blank_type = lab-only; use with caution").

All sites include both field and deionized water blanks; the only exception is OLI, where field blanks were not collected due to operational issues by the technicians at the site. In that case, laboratory blanks were used instead. Since this information would be largely redundant in the table, we instead stated it at the end of Section 3.2.1: "All sites undergo field blank collection approximately once per month, with the exception of OLI M1, which is detailed in a Data Quality Report (DQR) that can be viewed under the "Description" for this site's INP data on the ARM Data Discovery Portal. Moving forward, all existing and new sites in the program will include routine monthly field blanks."

In lieu of a flag, under "Description" for OLI after selecting "Details" and clicking on the timeline, we already note on ARM Data Discovery: "Blanks for the data set were shipped to but never opened in the field. They essentially represent laboratory cleanliness during filter unit preparations at Colorado State University. However, this likely does not affect the results, given the ARM operators are careful when handling filter units during collection at the site." See screen shot of the website below from https://adc.arm.gov/discovery/#/results/s::inp.

[Figure]

6.  TBS sampling details (Section 2.1.3, lines 144–152; Table 2): For TBS deployments, please document the standard conditions implied by the logged volumetric flow (Slpm) and any conversions applied to STP volumes at altitude. Include face velocity across the filter, residence time per filter/altitude, flow stability during ascents/descents, and the barometric/temperature corrections used. If these are detailed in Dexheimer et al. (2024), cross-reference explicitly.

Since this paper has undergone review, we have published a separate paper detailing the TBS system, its deployments, and associated data (Creamean et al., 2025). Consequently, we removed Table 2 and revised the final sentence of Section 2.1.3 to: "The TBS INP sampler design, filter preparations, deployments, and available data are described in detail in Creamean et al. (2025) and are only briefly mentioned here." We also removed Section 2.2.2, which focused on filter preparations for TBS deployments, as this is fully covered in our more recent paper. Information on how to access TBSINP data was retained, as it is essential to the available dataset and relevant to this data paper.

Creamean, J. M., Dexheimer, D., Hume, C. C., Vazquez, M., Hess, B. T. M., Longbottom, C. M., Ruiz, C. A., and Theisen, A. K.: Reaching new heights: A vertically-resolved ice nucleating particle sampler operating on Atmospheric Radiation Measurement (ARM) tethered balloon systems, EGUsphere, 1–22, https://doi.org/10.5194/egusphere-2025-5000, 2025.

B. Moderate suggestions

Positive control (Section 3.2.3, lines 272–283): Report periodic measurements of a standard INP material (e.g., Snomax, illite NX) to track sensitivity drift.

We do not use positive controls with a standard INP material to assess instrumental drift. Instead, potential drift or other issues (including those related to deionized water quality) are monitored through the deionized water blanks that are run alongside every sample. Thus the deionized water blanks serve a dual purpose. If a deionized water blank appears abnormal relative to expectations, we investigate potential issues by re-running the sample and water blank, evaluating possible sources of contamination, and/or checking whether INS components (i.e., thermocouples, cameras, etc.) may be malfunctioning. To make this clear, we added the following to Section 3.2.3: "The deionized water blanks run with every sample serve a dual purpose: they act as a positive control as well and help monitor instrument drift, potential contamination, and the proper functioning of INS components (e.g., thermocouples, cameras)."

Cross-site comparability (Figures 6–7; Section 4.3, lines 358–364): Daily sampling resolves episodic events better than 6-day routine sampling. Note this caveat and consider a sensitivity test subsampling daily periods to pseudo-6-day to quantify biases in seasonal boxplots.

See response to Comment 1. The samples are not 6-day integrations; rather, they are 24-hour samples collected approximately every 6 days. Therefore, episodic events (at least, those more than 24-hours) are already captured within these figures.

Treatment fractions (Section 2.3.1, line 224–225; Fig. 5, lines 342–346): State, per site and season, the fraction and number of samples undergoing heat and $H_2O_2$ treatments. Add treatment identifiers to records.

Due to the ongoing nature of this program, the numbers of treatments change continuously on a weekly basis. Readers can, however, view real-time statistics of the treatments at each site and the exact dates

when they were applied via the field log on our website. For example, the first entry for BNF as of 2 Oct 2025 shows totals for base (no treatments), heat, and peroxide at the top of the log. Because of this constant evolution, including treatment numbers in the manuscript would be outdated by the time of publication and thus not meaningful. To clarify this in the text, we added the following sentences to the end of Section 2.3.1, in the paragraph describing processing of one-third of the samples at most sites: "Due to the ongoing nature of this program, the numbers of treatments conducted on samples from any given site evolve continuously on a weekly basis. Exact sample sites, dates, identifiers, and other metadata regarding which samples undergo treatments can be accessed in real time via the publicly available field log on the INS website."

Treatment identifiers are already included in the records/data files as flags, as described in Section 3.1. We have added clarification in that section specifying the meaning of each flag: untreated = 0, heat treated = 1, and peroxide treated = 2.

| 52 | 40 | 40 | 8 | 8 | | | | | | | | nr = not recorded | nc = not collected | red = value was high | blue = value was low | | | | | | | |
|---|---|---|---|---|---|---|---|---|---|---|---|---|---|---|---|---|---|---|---|---|---|
| filter # | data proccessed | base | heat | H2O2 | checklist date | start (UTC) | end (UTC) | A start flow (slpm) | A start P (kPa) | A start T (C) | A end flow (slpm) | A end P (kPa) | A end T (C) | A total vol (L) | B start flow (slpm) | B start P (kPa) | B start T (C) | B end flow (slpm) | B end P (kPa) | B end T (C) | B total vol (L) |
| 1 | ☑ | ☑ | ☐ | ☐ | 10/16/24 | 10/01/24 14:05 | 10/02/24 14:05 | 14.46 | 47.80 | 22.8 | 11.87 | 42.93 | 23.90 | 1,998.90 | 14.70 | 46.90 | 24.4 | 12.25 | 39.95 | 23.8 | 19,940.8 |
| 2 | ☑ | ☑ | ☐ | ☐ | 10/16/24 | 10/07/24 14:03 | 10/08/24 14:03 | 13.70 | 48.59 | 23.3 | 12.29 | 43.44 | 22.80 | 19,976.00 | 14.61 | 50.57 | 24.1 | 14.11 | 42.33 | 23.2 | 22,948.5 |
| 3 | ☑ | ☑ | ☐ | ☐ | 10/16/24 | 10/15/24 14:09 | 10/16/24 13:58 | 13.12 | 54.96 | 22.9 | 13.58 | 45.97 | 23.40 | 20,433.60 | 12.32 | 54.64 | 23.4 | 13.14 | 42.29 | 24.1 | 19,893.5 |
| 4 | ☑ | ☑ | ☑ | ☑ | 10/22/24 | 10/21/24 14:07 | 10/22/24 14:02 | 14.56 | 53.65 | 19.7 | 12.13 | 44.32 | 21.90 | 21,592.60 | 13.99 | 53.47 | 20.4 | 9.93 | 38.77 | 22.7 | 20,315.0 |
| 5 | ☑ | ☑ | ☐ | ☐ | 10/28/24 | 10/27/24 14:01 | 10/28/24 13:57 | 12.74 | 55.62 | 20.9 | 8.52 | 39.77 | 23.60 | 17,544.00 | 15.80 | 55.06 | 21.5 | 10.25 | 38.96 | 24.0 | 21,314.9 |
| 6 | ☑ | ☑ | ☑ | ☑ | 11/04/24 | 11/02/24 14:10 | 11/03/24 14:10 | 13.03 | 49.08 | 23.2 | 7.01 | 37.81 | 22.90 | 15,854.50 | 14.83 | 51.61 | 23.9 | 9.29 | 38.34 | 23.5 | 18,920.7 |
| 7 | ☑ | ☑ | ☐ | ☐ | 11/21/24 | 11/08/24 14:54 | 11/09/24 15:00 | 14.68 | 47.95 | 21.0 | 10.82 | 43.08 | 23.10 | 18,955.80 | 15.36 | 46.46 | 21.6 | 11.55 | 40.58 | 23.5 | 20,146.0 |
| 8 | ☑ | ☑ | ☑ | ☑ | 11/21/24 | 11/14/24 14:55 | 11/15/24 14:55 | 9.27 | 51.34 | 21.1 | 9.44 | 41.48 | 22.90 | 14,192.50 | 13.58 | 54.75 | 21.7 | 13.58 | 43.15 | 22.6 | 21,331.7 |
| 9 | ☑ | ☑ | ☐ | ☐ | 11/21/24 | 11/20/24 14:59 | 11/21/24 15:00 | 8.68 | 47.90 | 21.1 | 9.60 | 42.45 | 22.50 | 13,966.20 | 14.66 | 51.00 | 21.6 | 15.35 | 45.60 | 22.3 | 22,593.3 |
| 10 | ☑ | ☑ | ☑ | ☑ | 12/03/24 | 12/02/24 14:46 | 12/03/24 14:56 | 9.21 | 50.55 | 22.3 | 9.05 | 43.36 | 19.90 | 13,905.60 | | | | | | | |
| 11 | ☑ | ☑ | ☑ | ☑ | 12/09/24 | 12/08/24 16:24 | 12/09/24 16:23 | 9.36 | 44.39 | 20.4 | 7.64 | 38.92 | 24.30 | 12,125.60 | 14.36 | 44.51 | 20.2 | 9.50 | 38.01 | 24.4 | 16,915.0 |
| 12 | ☑ | ☑ | ☐ | ☐ | 12/23/24 | 12/20/24 14:40 | 12/21/24 14:40 | 14.84 | 51.58 | 21.7 | 12.49 | 46.62 | 19.70 | 20,522.60 | 14.41 | 50.10 | 21.8 | 15.89 | 45.73 | 22.4 | 155.8 |

Software versioning (Section 3.3, line 286): Cite a DOI/Zenodo release or commit hash for OLAF DaQ INS and include version in dataset metadata.

We do not have a DOI or Zenodo release for OLAF, so we provided the GitHub link where it is already available as open-source software. We clarified this by revising the sentence in the sentence in Section 3.3 to: "More information and the software itself are available at: https://github.com/SiGran/OLAF."

C. Minor and editorial corrections

DOIs (lines 432 and 450): Several DOIs are written as https//doi.org/... (missing colon). Please correct to https://doi.org/....

Done.

Typo (Table 3): Remove duplicated word 'the the ARM G-1'.

Done.

Purge gas (line 213): Replace 'pre-cooled slightly above block temperature' with 'pre-cooled near block temperature'.

Done.

Figures: 4 (lines 318–330): Add number of samples per line.

Each line is based on the same number of samples from SGP, as these are data points from cumulative spectra. We now indicate in the figure caption that these data are from 247 total processed filter samples.

Fig. 5 (lines 342–346): Clarify number of treated samples.

We now indicate in the figure caption that these data are from 84 treated samples (34% of the processed SGP samples in Figure 4).

Figs. 6–7 (lines 358–364): Add sample counts per boxplot and note filter durations.

Done. All filter durations are 24-hour as indicated in the comments above.

Tables: Table 1: add typical duration per filter and blank type.

Added to footnote.

Table 2 (line 153): add residence-time assumptions.

See response to Comment #6.

I recommend "major revisions." The manuscript's core contribution is strong, but resolving sampling-duration ambiguity, fully documenting automated corrections/QC, and expanding the uncertainty budget are essential for a durable ESSD dataset paper.

**Reviewer #2**

This data description paper by Creamean et al. offers a bodacious ice-nucleating particle dataset. The manuscript is well-written and fulfills the journal's scope. This reviewer recommends publishing this paper for ESSD after the authors address the following comments.

The authors did a good job showing applications of their INP data. How about limitations? What do the data users need to be aware of when they apply the offered data in observation-driven Earth system models etc. on different spatial-temporal scales? Perhaps, the authors might address it according to the challenges discussed in Burrows et al. (2022; DOI - https://doi.org/10.1029/2021RG000745)? This reviewer believes that clearly stating limitations is as important as demonstrating the applicability of any data.

The reviewer brings up a very valid point. We changed the title of Section 5 to "Community use and limitations of ARM INP data" and have added the following text: "The DOE ARM baseline INP measurements provide valuable long-term and IOP-based observations but have several limitations that users should be aware of. First, these measurements do not account for time dependence in freezing behavior, which is generally less significant than temperature dependence (Ervens and Feingold, 2013). Second, sampling assumes collection of the total aerosol size distribution; however, this has not been

explicitly tested, so the exact size range collected is uncertain. The 0.2-µm filters we use have reduced transmission efficiency for particles around 150 nm (down to 65–78%), but generally exhibit very high collection efficiency at most sizes (Spurny and Lodge, 1972). Third, because filters are collected over 24 hours, typically every six days, short-term or episodic INP events may be missed, although higher-frequency sampling can be requested. Fourth, these measurements are made at the surface and may not fully represent the INP population at cloud level, though cloud-surface coupling analyses (e.g., Creamean et al., 2021; Griesche et al., 2021) and TBS INP data (Creamean et al., 2025) can help bridge this gap. Lastly, not all samples are subjected to treatments unless requested, and, as noted by Burrows et al. (2022), the absence of co-located baseline measurements of aerosol composition (particularly dust, sea spray, and primary biological aerosol particles) limits the ability to fully constrain INP sources and improve parameterizations in models."

L66-67: It seems controversial that the authors raise the concern of "not routine" INP measurements here, while they offer several single IOP data in this manuscript. The authors might want to rephrase this sentence; otherwise, clarify the concern rigorously.

Good point. What we intended to emphasize is that ARM now offers INP measurements as a baseline product. Accordingly, we revised the sentence to: "While INP measurements have been conducted at various ARM sites in the past, they were primarily user-driven and not part of the baseline measurement suite." We also changed "routine" on line 72 to "baseline."

L180-181: Has the time span between collection and analysis been consistent for all samples? If not, this reviewer would like to see if the authors can discuss the impact of various sample storage intervals.

The time between collection and analysis has varied from as little as one week to over a year. Beall et al. (2020) reported no significant differences in INP concentrations for samples stored between 1 and 166 days. For select samples re-analyzed as part of our internal quality checks, we observed minimal differences (<1%) after one to two and a half years of storage at −20 °C. In cases where re-analysis was required due to issues with the original measurements, differences of 20 to 60% were observed at some temperature bins; however, these represent outliers associated with problematic initial samples. We added this text to Section 2.3: "The time between collection and analysis has ranged from one week to over a year. Beall et al. (2020) reported no significant differences in INP concentrations for samples stored between 1 and 166 days. In our internal quality checks, select samples stored for 1–2.5 years at −20 °C showed minimal differences (<1%), while larger differences (20–60%) were observed only in outlier cases associated with problematic original measurements."

L220-221: What is the rational procedure of H2O2 application for organic removal? This reviewer is aware that the H2O2 concentration ranges (e.g., Perkins et al., 2020: DOI - 10.1021/acsearthspacechem.9b00304). It seems the peroxide digestion method is operational without verification. The authors might consider including a brief yet clear statement of what needs to be investigated in terms of H2O2 application down the road in the manuscript. Doing this will benefit the community and mitigate the concerns of readers.

The use of $H_2O_2$ for degrading organic matter in soils dates back to Robinson (1922), who first introduced it for soil texture analysis (Mikutta et al., 2005). Since then, it has become the most widely used chemical

reagent for organic matter destruction prior to textural and mineralogical analyses. Typical treatments employ 30% (w/w) $H_2O_2$, though concentrations ranging from 6% to 50% have been used with little difference in carbon removal efficiency (McLean, 1931), likely due to rapid $H_2O_2$ decomposition and the persistence of mineral-protected organic compounds. Early protocols began at room temperature to accommodate vigorous reactions with easily decomposable material, followed by heating (60–90 °C) to accelerate oxidation (Schultz et al., 1999). Despite its century-long use, there remains no standardized protocol for $H_2O_2$ treatment in soil science, with reported concentrations, reaction temperatures, and contact times varying widely depending on research objectives and soil properties (e.g., Sequi and Aringhieri, 1977).

In the presence of UV, $H_2O_2$ photolyzes, producing hydroxyl radicals ($H_2O_2$ + hv (UV) → 2 ·OH). The resulting hydroxyl radicals are highly reactive and can attack functional groups on polysaccharides, open sugar rings, and oxidize carbonyl (–C=O) or carboxylic acid (–COOH) groups (e.g., Chen et al., 2021; Ofoedu et al., 2021). Such reactions can disrupt molecular integrity and crystallographic order, oxidize or rearrange hydrophilic groups, and alter polymer conformation (e.g., through unwinding, depolymerization, or cross-linking). These structural modifications can diminish ice-nucleating activity of bio-organic materials by reducing the number of effective ice-active sites, or, if extensive enough, completely eliminate activity by destroying the structural motifs responsible for ice templating (DeMott et al., 2023; Gute and Abbatt, 2018).

Within the INP community, treatments with 10% $H_2O_2$ are most commonly used, although concentrations ranging from 10% to 35% have been reported (e.g., Barry et al., 2023a; Beall et al., 2022; Hill et al., 2016; McCluskey et al., 2018; O'Sullivan et al., 2014; Roy et al., 2021; Suski et al., 2018; Tesla et al., 2024; Testa et al., 2021; Tobo et al., 2019). Perkins et al. (2020) used 15% $H_2O_2$ and heated to 95 °C, but organics in their ATD tests were found to be thermally stable to 500 °C without peroxide digestion. In our recent tests comparing 10% and 20% treatments on the same samples, we observed minimal differences in ice nucleating activity; however, additional experiments are needed to confirm this result.

As suggested by the reviewer, we have provided this text to the section 2.3.1: "$H_2O_2$ has long been used as an oxidizing agent for degrading organic matter, originating in soil science nearly a century ago and later adopted across disciplines for removing organic material prior to chemical or physical analyses (McLean, 1931; Mikutta et al., 2005; Robinson, 1922; Schultz et al., 1999; Sequi and Aringhieri, 1977). In the presence of UV light, $H_2O_2$ photolyzes to form highly reactive hydroxyl radicals that can oxidize and structurally modify organic macromolecules, diminishing or eliminating their ice-nucleating activity (DeMott et al., 2023; Gute and Abbatt, 2020). Within the INP community, $H_2O_2$ treatments typically range from 10 to 35% (Beall et al., 2022; O'Sullivan et al., 2014; Perkins et al., 2020; Roy et al., 2021; Teska et al., 2024; Tobo et al., 2019). We conducted recent tests showing minimal differences in ice-nucleating activity between 10% and 20% treatments; however, further validation is needed, and future community efforts should aim to establish a standardized protocol and concentration to ensure methodological consistency across studies."

References:

Beall, C. M., Hill, T. C. J., DeMott, P. J., Köneman, T., Pikridas, M., Drewnick, F., Harder, H., Pöhlker, C., Lelieveld, J., Weber, B., Iakovides, M., Prokeš, R., Sciare, J., Andreae, M. O., Stokes, M. D., and

Prather, K. A.: Ice-nucleating particles near two major dust source regions, Atmospheric Chemistry and Physics, 22, 12607–12627, https://doi.org/10.5194/acp-22-12607-2022, 2022.

Chen, X., Sun-Waterhouse, D., Yao, W., Li, X., Zhao, M., and You, L.: Free radical-mediated degradation of polysaccharides: Mechanism of free radical formation and degradation, influence factors and product properties, Food Chemistry, 365, 130524, https://doi.org/10.1016/j.foodchem.2021.130524, 2021.

DeMott, P. J., Hill, T. C. J., Moore, K. A., Perkins, R. J., Mael, L. E., Busse, H. L., Lee, H., Kaluarachchi, C. P., Mayer, K. J., Sauer, J. S., Mitts, B. A., Tivanski, A. V, Grassian, V. H., Cappa, C. D., Bertram, T. H., and Prather, K. A.: Atmospheric oxidation impact on sea spray produced ice nucleating particles, Environ. Sci.: Atmos., 3, 1513–1532, https://doi.org/10.1039/D3EA00060E, 2023.

Gute, E. and Abbatt, J. P. D.: Oxidative Processing Lowers the Ice Nucleation Activity of Birch and Alder Pollen, Geophys. Res. Lett., 45, 1647–1653, https://doi.org/10.1002/2017GL076357, 2018.

McLean, W.: The Nature of Soil Organic Matter as Shown by the attack of Hydrogen Peroxide, The Journal of Agricultural Science, 21, 595–611, https://doi.org/10.1017/S0021859600009813, 1931.

Mikutta, R., Kleber, M., Kaiser, K., and Jahn, R.: Review: Organic Matter Removal from Soils using Hydrogen Peroxide, Sodium Hypochlorite, and Disodium Peroxodisulfate, Soil Science Society of America Journal, 69, 120–135, https://doi.org/10.2136/sssaj2005.0120, 2005.

O'Sullivan, D., Murray, B. J., Malkin, T. L., Whale, T. F., Umo, N. S., Atkinson, J. D., Price, H. C., Baustian, K. J., Browse, J., and Webb, M. E.: Ice nucleation by fertile soil dusts: relative importance of mineral and biogenic components, Atmospheric Chemistry and Physics, 14, 1853–1867, https://doi.org/10.5194/acp-14-1853-2014, 2014.

Ofoedu, C. E., You, L., Osuji, C. M., Iwouno, J. O., Kabuo, N. O., Ojukwu, M., Agunwah, I. M., Chacha, J. S., Muobike, O. P., Agunbiade, A. O., Sardo, G., Bono, G., Okpala, C. O. R., and Korzeniowska, M.: Hydrogen Peroxide Effects on Natural-Sourced Polysaccharides: Free Radical Formation/Production, Degradation Process, and Reaction Mechanism—A Critical Synopsis, Foods, 10, 699, https://doi.org/10.3390/foods10040699, 2021.

Robinson, G. W.: Note on the mechanical analysis of humus soils, The Journal of Agricultural Science, 12, 287–291, https://doi.org/10.1017/S0021859600005347, 1922.

Roy, P., House, M., and Dutcher, C.: A Microfluidic Device for Automated High Throughput Detection of Ice Nucleation of Snomax®, Micromachines, 12, 296, https://doi.org/10.3390/mi12030296, 2021.

Schultz, M. K., Biegalski, S. R., Inn, K. G. W., Yu, L., Burnett, W. C., Thomas, J. L. W., and Smith, G. E.: Optimizing the removal of carbon phases in soils and sediments for sequential chemical extractions by coulometry, J. Environ. Monit., 1, 183–190, https://doi.org/10.1039/A900534J, 1999.

Sequi, P. and Aringhieri, R.: Destruction of Organic Matter by Hydrogen Peroxide in the Presence of Pyrophosphate and Its Effect on Soil Specific Surface Area, Soil Science Society of America Journal, 41, 340–342, https://doi.org/10.2136/sssaj1977.03615995004100020033x, 1977.

Teska, C. J., Dieser, M., and Foreman, C. M.: Clothing Textiles as Carriers of Biological Ice Nucleation Active Particles, Environ. Sci. Technol., 58, 6305–6312, https://doi.org/10.1021/acs.est.3c09600, 2024.

Tobo, Y., Adachi, K., DeMott, P. J., Hill, T. C. J., Hamilton, D. S., Mahowald, N. M., Nagatsuka, N., Ohata, S., Uetake, J., Kondo, Y., and Koike, M.: Glacially sourced dust as a potentially significant source of ice nucleating particles, Nat. Geosci., 12, 253–258, https://doi.org/10.1038/s41561-019-0314-x, 2019.

L232: Missing a negative sign on RHS in Eqn (1)?

Thanks for catching that typo. We have fixed it.

L233: How do the authors determine V_suspension values? Please clarify in the manuscript.

We have clarified this in Section 3.1: "Specifically, the $V_{suspension}$ is the volume of $0.1\,\mu$m–filtered deionized water used to resuspend the particles from the filter (7–10 mL)."

---

## Author Response (AR2)

We thank the editor for catching what we had missed in our response to the reviewers. Our detailed responses to the editor comments are provided below in blue.

Dear Authors,

The manuscript is suitable for publication pending minor revisions. Please address the following points, and provide justifications if any cannot be implemented:

Reviewer #2 comment on "Automated monotonicity correction (Section 3.3, lines 297–302; Fig. 3)": The clarification regarding error flags and future QC flag implementation is appreciated. However, the response does not answer the reviewer's request to quantify the frequency of monotonicity corrections or include a counter. Please provide a summary statistic or typical range per sample/site, or explain why this cannot be included, to improve transparency and allow downstream users to understand how often corrections are applied.

Thank you for noticing that we did not fully address Reviewer #2's comment. In Section 3.3, we have added the following text:

"In cases where INP concentrations decrease with decreasing temperature, an artifact sometimes introduced by the stochastic nature of the measurement produced by multiple serial dilutions, the program automatically adjusts the value to maintain monotonicity using a two-fold check. Blank subtraction can also introduce this artifact; therefore the correction is applied after the blank subtraction.

First, a filter is applied to ensure that values genuinely affected by blank subtraction are not included in the monotonicity correction. If a blank-corrected value falls below the lower 95% confidence bound of the uncorrected value, the program replaces it with the previous bin's value and propagates the upper confidence interval using the root mean square of the current and previous intervals. The lower confidence bound from the previous value is applied to the current value. This first correction is applied only if occurrences remain below a user-defined threshold (10% of total temperature bins per sample or approximately 4 temperature bins). If exceeded, the affected bins are flagged with an error signal (–9999).

Then, the monotonicity check is performed on the filtered values. If a filtered value decreases from the value in the previous temperature bin, the program replaces it with the previous previous bin's value and propagates the upper confidence interval using the root mean square of the current and previous intervals. The lower confidence bound from the previous value is applied to the current value.

Thus far, OLAF has only been used to process four sets of data that are available on Data Discovery from TBS deployments in 2025. Of those sets, the monotonicity correction was applied to 68% of the samples, on average correcting less than two temperature bins per sample. The correction was applied almost completely due to dilution stochasticity and rarely due to blank subtraction. OLAF will be used to generate data from INS processing at all sites moving forward. We expect ground-based sites to be similar or experience less frequent corrections due to higher collection volumes. Finally, the software compiles the blank-corrected data across all treatments (base, heat, and peroxide) into a single output file, including treatment flags for each sample."

Reviewer #2 comment on "Blank strategy and Oliktok (Sections 2.1.2–2.1.3, Table 1; Section 3.2.1, lines 254–255)": While the ARM portal description for Oliktok (OLI) is helpful, the reviewer's intent was to ensure that blank type is explicitly encoded in the data product and easily discoverable by users without consulting external documentation. Including a metadata field (e.g., blank_type) or a brief note in Table 1 (e.g., "OLI = lab-only") would strengthen transparency and reproducibility. Please revise accordingly or provide a clear rationale if this cannot be implemented in the current data release.

Since submitting the revision, all coauthors have discussed and agreed to add new metadata fields to enhance the dataset's contextual information. Specifically, we are reprocessing all data files to include two new global metadata attributes in the NetCDF files: sample_notes and filter_color. The sample_notes field will contain relevant field observations provided in our field log and indicate cases where laboratory blanks were used (as for OLI), while filter_color will indicate the observed filter color upon collection. These additions are now noted in Section 3.4 and reiterated in Section 4.3, where the OLI blanks are discussed.

Reviewer #2 comment on "Software versioning (Section 3.3, line 286)": You have provided the GitHub link for OLAF. Please also specify the current commit hash or version tag in the manuscript and ensure this version identifier is included in the dataset metadata. If a DOI (e.g., via Zenodo) becomes available before publication, please include it. Specifying the commit or version is important to ensure reproducibility and traceability, allowing future users to know exactly which software version was used to generate the reported data.

We were able to create a DOI for the current version and have now included it in the manuscript in Section 3.3 (https://doi.org/10.5281/zenodo.17509699). This leads one to OLAF v0.3.0-beta release.

Once these revisions have been addressed or appropriately justified, the manuscript can be accepted for publication.